

**An overview on the airborne measurement in Nepal,-part 1: vertical profile of aerosol size-number,**
**spectral absorption and meteorology**
Ashish Singh[1]*, Khadak S. Mahata[1], Maheswar Ruphaketi[1]*, Wolfgang Junkermann[2], Arnico K. Panday[3],
Mark G. Lawrence[1]
[1]Institute for Advanced Sustainability Studies, Potsdam, Germany
[2]Institute of Meteorology and Climate Research, IMK-IFU, Garmisch-Partenkirchen, Germany
[3]International Centre for Integrated Mountain Development (ICIMOD), Lalitpur, Nepal
*Corresponding author: Ashish Singh (ashish.singh@iass-potsdam.de) and
Maheswar Rupakheti (maheswar.rupakheti@iass-potsdam.de)



**Abstract**

14        The paper provides an overview of an airborne measurement campaign with a microlight aircraft,

over the Pokhara Valley region, Nepal, a metropolitan region in the central Himalayan foothills. This is
the first aerial measurements in the central Himalayan foothill region, one of the polluted but relatively
poorly sampled regions of the world. Conducted in two phases (in May 2016 and December 2016-
January 2017), the goal of the overall campaign was to quantify the vertical distribution of aerosols over
a polluted mountain valley in the Himalayan foothills, as well as to investigate the extent of regional
transport of emissions into the Himalayas. This paper summarizes results from first phase where test
flights were conducted in May 2016 (pre-monsoon), with the objective of demonstrating the potential of
airborne measurements in the region using a portable instrument package (size with housing case: 0.45
m x 0.25 m x 0.25 m, 15 kgs) onboard an ultralight aircraft (IKARUS-C42). The limited dataset collected
during the test flight also provides useful insights into the impact of regional emissions and meteorology
on aerosol vertical profiles. A total of five sampling test flights were conducted (each lasting for 1-1.5 h)
in the Pokhara Valley to characterize vertical profiles of aerosol properties such as aerosol number and
size distribution (0.3-2 μm), total particle concentration (>14 nm), aerosol absorption (370-950 nm),
black carbon (BC), and meteorological variables.

The vertical profiles of aerosol species showed decreasing concentrations with altitude (815 to

4500 m a.s.l.); steep concentration gradient below 2000 m (a.s.l.) in the morning and a more mixed
profiles (up to ca. 4000 m a.s.l.) in the afternoon. The strong gradient in the morning hours was mainly
contributed by the primary emissions from the valley floor, including occasional open agriculture
burning. The near-surface (<1000 m a.s.l.) BC concentrations observed in the Pokhara Valley were much
lower than pre-monsoon BC concentrations in the Kathmandu Valley, and similar in range to Indo-
Gangetic Plain (IGP) sites such as Kanpur in India.  The sampling test flight also detected an elevated
polluted aerosol layer (around 3000 m a.s.l.) over the Pokhara Valley, which could be associated with
the regional transport. The total aerosol and black carbon concentration in the polluted layer was
comparable with the near-surface values (<1000 m a.s.l.). The elevated polluted layer was also
characterized by high aerosol extinction co-efficient (at 550 nm) and was identified as smoke and a
polluted dust layer. Long-term observations of aerosol optical depth (AOD) in the Pokhara Valley (2010-
16) showed strong seasonality, with a pre-monsoon maximum which is also indicative of westerly
advection transporting a mixture of dust and other aerosols from IGP into Himalayan foothills and
mountain valleys. The observed shift in the westerlies (at 20-30° N) entering Nepal during the test flight



44 period is an important factor for the presence of elevated polluted layers in the Pokhara Valley.  The

45 intrusions (in the form of a trough) of the cold and humid air mass from the mid-latitude (~ 40-50° N) a

46 shift in the direction of synoptic airmass entering Himalayas. This synoptic-scale interaction is likely to

47 drive the transport into the mountain valleys and higher Himalayas.

48 **1. Introduction**

49  The Himalayas and surrounding regions are one of the unique ecosystems in the world, with a great

50 variety in the geography and socio-economics, and a notable significance in the context of regional and

51 global environmental change.  Areas in the foothills of the Himalayas still constitute large regions of

52 rural populations along with pockets of rapidly growing cities. Consequently, there is a complex

53 interaction among changing emission sources and their interaction with regional and global climate

54 change. Among emitted air pollutants, the chemical and physical properties of aerosols have been linked

55 to significant burdens of disease, to melting of glaciers, to crop loses, to hydrological changes and to

56 cloud properties (Bollasina et al., 2011;Vinoj et al., 2014;Lau, 2014;Burney and Ramanathan,

57 2014;Brauer et al., 2012;Cong et al., 2015;Li et al., 2016).

58  Sources of aerosols in the Himalayas and the nearby Indo-Gangetic Plain (IGP) typically vary

59 between urban, peri-urban and rural locations; fossil fuel and industrial emissions such as vehicles, brick

60 kilns, waste burning, cement factories etc., are typically urban and peri-urban; biomass cookstove,

61 agriculture and waste burning and forest fires are often linked to emissions from rural areas (Guttikunda

62 et al., 2014;Venkataraman et al., 2006;Stone et al., 2010). Secondary chemical pathways also contribute

63 to the aerosols in the ultrafine and accumulation-mode range via particle formation events (Venzac et

64 al., 2008).

65  Aerosol properties in the Himalayas have large spatial and temporal variations, especially in the pre-

66 monsoon and monsoon season. These observed variations are influenced by emission sources, regional

67 meteorology and geography (Dey and Di Girolamo, 2010). The influence of aerosol particles on local and

68 regional weather during these adjacent seasons has significant implications for timing, intensity and

69 spatial distribution of the summer monsoon in the region (Bollasina et al., 2011;Ramanathan et al.,

70 2001). Studies describing the aerosol-meteorology interaction are often missing in the Himalayan region

71 partly due to lack of surface and airborne measurements of aerosol properties along with meteorology.

72 Most past campaign-mode measurements in the Himalayan regions, to our knowledge, have been

73 ground measurements, which have aided in evaluating aerosol properties, and their transformation and

74 transport mechanisms (Shrestha et al., 2013;Shrestha et al., 2010;Ramana et al., 2004;Marcq et al.,

75 2010;Panday and Prinn, 2009;Cho et al., 2017). Long-term continuous measurements of aerosols and





meteorology are limited to a few stations in the High Himalayas, such as the recently discontinued Nepal
Climate Observatory at Pyramid (NCO-P, 27.95° N, 86.81° E, 5050 m a.s.l.), a high altitude observatory
located near basecamp of Mt. Everest. Columnar and satellite measurements such as AERONET and
CALIPSO have provided a regional overview of aerosol type and vertical distribution, as well as
estimation of aerosol heating rate in the atmospheric column (Kuhlmann and Quaas, 2010;Gautam et
al., 2011;Pandey et al., 2017).However, these measurement techniques often suffer from large
uncertainty and biases while retrieving the complex nature of the aerosols observed in the region (Jai
Devi et al., 2011).

Regional meteorology in the 850-500 mb range plays an important role in the transformation and

transport of aerosols from Western Asia to the IGP, the Himalayan foothills, the Himalayan and Tibetan
Plateau region (Decesari et al., 2010;Marinoni et al., 2013;Lüthi et al., 2015). At these altitudes,
synoptic- scale air masses are mostly westerly/northwesterly during the pre-monsoon and
southwesterly/easterly during the monsoon. These air masses are often linked to dust aerosol transport
during the pre-monsoon season from Western Asia into the Himalayas, including populated mountain
valley such as Kathmandu and Pokhara Valley in Nepal. The transported dust aerosol also mixes with the
primary emission (or anthropogenic aerosols) in the IGP and accumulates from northern to eastern IGP
along the Himalayan foothills (Gautam et al., 2009b;Gautam et al., 2011). The total aerosol loading is
often highest during the pre-monsoon season in the IGP (Gautam et al., 2009a;Raatikainen et al., 2014),
intensified further by weak surface/zonal winds and numerous open biomass burning and forest fires
events (Kaskaoutis et al., 2012b). The polluted aerosol layer in the IGP is advected into the Himalayas by
synoptic-scale westerlies (~500 mb) and also by the valley wind circulation within or along the planetary
boundary layer (PBL) (Lüthi et al., 2015). The advection is also facilitated by strong updraft and PBL
expansion (highest in the pre-monsoon in the IGP) often mixing with the synoptic-scale westerlies
(Raatikainen et al., 2014).  Because of strong convective activity in the IGP, the polluted air masses near
the surface are often lifted up to 5-7 km or higher (Kuhlmann and Quaas, 2010). In addition to the
synoptic-scale transport, thermally-driven valley winds also enable the transport of humid and polluted
air mass (with enhanced absorbing fraction) from IGP into the Himalayan foothills, and further up into
the mountain valleys and elevated locations (Raatikainen et al., 2014;Lüthi et al., 2015;Gogoi et al.,
2014;Putero et al., 2014;Decesari et al., 2010;Marcq et al., 2010). Strongly coupled with the expansion
of the PBL in the IGP, the upslope movement of polluted air masses into the foothills and further east is
characterized by late afternoon peaks in AOD many measurement sites along the Himalayan range such
as Hanle Valley (Ladakh, India), Mukteswar and Manora site (Nainital, India), Hetauda (Nepal), Langtang





Valley (Nepal), Dhulikhel (Nepal), Kathmandu Valley (Nepal) and NCO-P (Nepal). The temporal and
spatial extent of this observed "ventilation" at multiple locations could be indicative of a regional-scale
transport than mesoscale (Gogoi et al., 2014;Raatikainen et al., 2014;Gautam et al., 2011;Putero et al.,
2015;Marcq et al., 2010).
To date there have been no observations of vertical distributions of aerosol and gaseous species
carried out in the Himalayan region. Therefore, the airborne measurement campaign was designed to
address two major questions: (i) what is the variation in the aerosol properties, notably the vertical
distributions, over a polluted mountain valley, and (ii) what is the quantitative extent of regional
transport of aerosols in the higher Himalayas? The campaign was carried out in two phases in the
Pokhara Valley and surrounding areas in Nepal.  In the first phase, test flights were conducted in May
2016 and in the second phase, intensive sampling flights were carried out in December 2016-January
2017. This paper provides an overview of the measurement campaign and results from the test flights in
May 2016 which include snapshots of vertical profiles of aerosol size, number, and composition, along
with meteorological parameters. The airborne measurements presented in this paper are supplemented
with observations of local and regional meteorology, as well as satellite and ground-based column-
integrated aerosol microphysics and radiative properties (see section 3.1.1 and 3.1.2). A companion
paper will follow with more detailed observations and results based on the intensive measurements
carried out during December 2016-January 2017.

**2.   Ultralight measurements in Nepal**
*2.1.  Details of airborne measurement unit*
A single-engine two-seater microlight aircraft (IKARUS C-42, COMCO IKARUS, Germany) was used as
the aerial platform. The technical specification of the aircraft includes approximately 4 h of flying time, a
short take-off run, an additional payload of up to 50 kg, and is suitable for spiral movement in the air.
The aircraft has a cruising speed of 165 kmh$^{-1}$, and a 5-6 ms$^{-1}$ rate of climb which makes it an appropriate
aerial vehicle to perform measurements at altitudes within the PBL and as close as 50 m above ground
level. More detail about the aircraft is available here (http://www.comco-ikarus.de/Pages/c42a-
technik.php?lang=en). Its size, speed and maneuverability offered a decent climb to the free
troposphere to capture vertical profiles in the rough terrains of Nepal. The aircraft used for the study is
operated by the Pokhara Ultralight Company for recreational flights around the Pokhara Valley.



The instrument package was specifically designed and tested for aerial measurements (Junkermann,
2001). Table 1 describes each instrument and the integration performed to prepare the package for the
aerial deployment.  The instrument package consists of a GRIMM 1.108 for particle size distributions
(0.3 to 20 μm, 16 size bins)  with sampling frequency of 6 s, and a TSI CPC 3760 for total particle
concentration (>11 nm) at 1 s resolution (See Figure S1 in the supplement). The package also included a
Magee Scientific aethalometer (AE42) for aerosol absorption at seven different wavelengths (370 -1020
nm).  The instruments were reduced in weight for use on the aircraft. The CPC was operated with a
constant mass flow and an internal DC pump instead of the original flow regulation by a critical orifice.
Meteorological parameters including temperature and dew point were sampled at a rate of 1 s or
higher. All the sensors were connected to a modular computer (PC104) for data acquisition. The PC104
is also equipped with a Global Positioning System (GPS), and multiple serial and analog connectors. For
inflight instrument checks and quick online overview of the atmospheric conditions, a small LCD was
also connected to the PC104 and placed in the cockpit areas for the flight crew. This display showed
real-time aerosol number concentrations and meteorological parameters.
**Table 1. Instrument package deployed in the microlight aircraft**
The instrument package weighs approximately 15 kg and consumes <60 W, well within the
power supply range of the aircraft battery. It is housed in an aluminum box (0.45 m x0.25 mx0.25 m),
and can be easily integrated with a mobile platform such as the IKARUS (See Figure S1). In IKARUS, the
instrument was placed in the rear section behind the seats which is otherwise almost empty, and only
contains the fuel tank and supporting aluminum bars. The sample inlet line (internal diameter of 0.004
m or ~4.0 mm ID brass tubing) ran along the wingspan and was approximately 1.8 m from the cockpit.
Once the sample line is inside the aircraft, it is distributed to all the aerosol instruments using a simple
metal flow splitter (0.006 m ID).  The sample inlet positioning at the end of the wingspan also minimizes
the influence of the aircraft propeller, located in the front of the cockpit.
*2.2.    Site description*
Pokhara Valley is Nepal's second largest populated valley (pop. >250,000) after the Kathmandu
Valley (CBS, 2011).  The valley is approximately at 815 m (a.s.l.), ~150 km west of the Kathmandu Valley,
and ~90 km northeast of the southern plains (~100 m a.s.l.) bordering IGP. The valley is surrounded by
mountains which are approximately 1000-2000 m (a.s.l.). Further north of the Pokhara Valley, within 30
km the elevation gradient increases rapidly to over 7000 m (a.s.l.) or higher (see Figure 1). This steep



elevation gradient is conducive for the orographic lift of humid air masses, and thus the valley also
receives one of the highest rates of precipitation in Nepal and occasional strong convective updrafts
leading to hailstorms and thunderstorms (Aryal et al., 2015). The mixing of dry westerly air masses with
heated moist air masses from the Bay of Bengal produces strong convection over the Pokhara Valley,
and thus results in strong updrafts. These strong convective activities are frequent in the pre-monsoon
and monsoon season, but do not occur during the winter season.
*2.3. Test flight patterns over the Pokhara Valley*
Five test flights were conducted in the morning and evening period around Pokhara Valley (83.97°
E, 28.19° N, 815 m a.s.l.) with each flight lasting for about 1 to 1.5 h from 5-7[th] May 2016. The flight
pattern was consistently flown over the northwest part of the valley (Figure 1).  A typical flight would
commence from the Pokhara Regional Airport (818 m a.s.l.) and steadily fly 5-10 km northwest along the
Pokhara Valley toward the Himalayas. This was followed by the spiral up and down sampling from
approximately 1000 to 4000 m, often reaching close to the lower base of the cloud in the free
troposphere. Further climbs into the cloud layer were avoided during the test flights.

**Figure 1.**  A typical test flight within the Pokhara Valley on 5[th] May 2017.  The plot is generated using a
Matlab-Google Earth toolbox (https://www.mathworks.com/matlabcentral/fileexchange/12954-google-
earth-toolbox). Each dot is a single sample point (sampling frequency of 1Hz); the color of the dot
indicates the total aerosol concentration and the value of each color is shown as a color bar.
*2.4. Data processing and quality*
The data from all the instruments is synced with the GPS clock, and the PC104 receives all the data
simultaneously and creates a common time-stamped data file. Prior to each test flight, a zero test was
conducted to identify any possible leaks in the sample line.
The collected data from the five test flights went through multiple steps of cleaning and flagging.
Occasionally during the radio communication by the pilot with the ground station or air traffic
controller, the CPC and the temperature sensor would record exceedingly high values.  This noise is an
interference picked up by the sensor from the 5 W radio transmission. The CPC and aethalometer is also
sensitive to vibration in the aircraft, especially during upward and downward spiral motion, which may
result in flow imbalance in these analyzers. This resulted in random noise segments for few seconds in
the data, which were flagged.



## 3. Results

*3.1. General meteorology and air quality, aerosol properties in the Pokhara Valley*

*3.1.1. Local and synoptic meteorology in the Pokhara Valley*

Climatologically, Pokhara Valley has a humid subtropical climate, characterized by a summer monsoon season from late June to September, preceded by a dry pre-monsoon (March-May, see Figure S2 in the supplement). Average monthly values of commonly measured meteorological parameters, shown in Figure S2 for 2016, are similar to the in range and variations observed in other studies (Poudyal et al., 2014;Khadka and Ramanathan, 2013). The annual mean temperature in the valley was 22° C, with the lowest monthly mean in January (~15° C) and the highest in July (~ 25° C). Rainfall was also highest in the months of August and September (summer monsoon season), followed by relatively dry post-monsoon (October-November) and winter period (December-February).  The late pre-monsoon to summer monsoon were also the periods of maximum monthly solar insolation (~900 Wm$^{-2}$) and the insolation is approximately half (~550 Wm$^{-2}$) during the winter. The dominant local/surface winds in the Pokhara Valley were from southeast and southwest followed by the northwest. The wind speed has a strong diurnal variability in the valley (Aryal et al., 2015) with low wind speed (<2.0 ms$^{-1}$) before noon-time, usually from southeast, followed by stronger winds from the southwest and northwest (>2.4 ms$^{-1}$) which can continue until late night. The increased wind speed in the afternoon could be katabatic in nature as  a result of differential heating in the mountain valley slopes and could be linked to pollution transport from surrounding regions (Gautam et al., 2011).  Winds in May 2016 were predominantly from the southeast with occasional strong winds from the southwest (see Figure S3 in the supplement). During the test flight period (5-7$^{th}$ May 2016), the wind was similar in directionality, with an hourly mean wind speed of 1.8 to 3.0 ms$^{-1}$.

Three dominant synoptic meteorology regimes characterize the seasonality of South Asia (Lawrence and Lelieveld, 2010). They are summer (June-September), the winter monsoon (mid-November to February) and the monsoon-transition periods, which include the pre-monsoon season (March-May) and post-monsoon season (mid-September to mid-November). These synoptic regimes are also active in the Himalayas, including the Pokhara Valley.  The monsoon transition period, during which the test flights measurements were conducted, is characterized by westerlies over 20-30° N at 850 mb and above (see Figure 2). Figure 2 shows the daily wind vector over South Asia for 3, 5, 6 and 7 May 2016 generated using the NCEP NCAR Reanalysis data at 2.5°x 2.5° horizontal resolution. While the reanalysis data can be expected to represent the synoptic-scale phenomena in this region reasonably



well, for the rough terrain in the Himalayas presents a significant challenge for modelling and the data is
thus likely to suffer from biases and other deviations from the observed meteorology (Xie et al., 2007).
The wind vector at 850 mb in the 20-30° N latitude band was westerly with variable wind speeds in the
IGP region near the Himalayan foothills. The wind direction varies diurnally at the 850 mb level, with the
wind direction shifting to southwesterly near the Himalayan foothills.  Westerlies were also generally
prevalent at the 500 mb; however, in the mid-latitudes between 40-50° N (Central Asia) , a trough and
crest-like feature of the westerlies moving from west to east Asia is visible (also observed by Lüthi et al.,
2015), which was also present prior to the study period. This wind feature was colder and more humid
(see Figure S4 in the supplementary material) than the westerlies observed between 20-30° N.  The
meandering features (i.e., trough and crest) observed between 40-50° N affects the direction and
magnitude of air masses (at 20-30° N) entering Nepal. For instance, the crest feature of the westerly was
prevalent over the IGP and Nepal prior to 3$^{rd}$ May, transitions into the trough feature after the 3$^{rd}$ and
continues during the study period. The prevalence of the trough was characterized by the intrusion of
wind into lower latitudes as well as into the IGP, also indicated by the change in the temperature and
humidity (Figure S4). The intrusions of mid-latitude air masses also influence the westerlies entering
Nepal in the 20-30° N sector (Lüthi et al., 2015).  As discussed later, variations in the vertical profiles of
aerosols above 3000 m (a.s.l.) could be associated with variations observed in these upper layer winds.

**Figure 2.**  Daily wind vector data at 850 and 500 mb, plotted using the NCEP NCAR reanalysis (2.5° x 2.5°)
data over South Asia from 1-7$^{th}$ May 2016. The colors indicate the wind speed in ms$^{-1}$. The plots were
generated using the default setup at www.esrl.noaa.gov/psd/data/composites/day/.

*3.1.2.    Summary of aerosol properties using AERONET measurement from 2010-2016*

AERONET measurements (Holben et al., 1998) have been made in the Pokhara Valley since January

2010.  The AERONET station (83.97° N, 28.18° E, 807 m a.s.l.) is approximately 1.1 km southeast from
the Pokhara Regional Airport, located in the semi-urban area of Pokhara City. Cloud-screened and
quality assured (level 2) data were used in the study. Gaps in the level 2 data were supplemented with
level 1.5 data. The AERONET retrieval suffers in the monsoon months (June to September) due to
interference by monsoon clouds in the Pokhara Valley, as indicated by the gap in Figure 2.

A combination of direct products such as aerosol optical depth (AOD) and inversion products such

as fine AOD, absorption Ångstrom exponent (AAE) and volume size distribution were used for the
analysis presented in this study. The typical reported uncertainty in the AERONET data products for AOD



(> 0.04) is approximately ±0.01 to ±0.02, and is higher for shorter wavelengths (Eck et al., 1999;Holben
et al., 1998).  The observed uncertainty in AOD also influences other AERONET products such as the
Ångstrom exponent (AE) and the inversion products. Thus these derived products will have a higher
uncertainty than the AOD (Schuster et al., 2006;Dubovik and King, 2000). Further details about the
AERONET direct and inverted data products can be found in Holben et al. (2006).
AOD and ground-level PM generally correlate well (Green et al., 2009), although the strength of
this association is greater with $PM_{2.5}$ (particulate matter less than 2.5 µm in aerodynamic diameter) than
with $PM_{10}$ (particulate matter less than 10 µm in aerodynamic diameter), and is also greater at moderate
RH levels than in moist air. The association between $PM_{10}$ (particulate matter less than 10 µm in
aerodynamic diameter) and AOD might suffer from interference due to the mixed nature of aerosol
particles, complex and changing sources of aerosols, and variable meteorological conditions (Singh et al.,

2004).

In the Pokhara Valley, AOD values showed a strong seasonality in the wavelength bands between
340 and 1020 nm.  The inter-annual variation in the AOD during 2010-2016 was closely associated with
the enhancement in the fine-mode fraction, and to a lesser extent in the coarse mode for dust (Xu et al.,
2014). The observed inter-annual variation in the AOD could be influenced by the interaction between
aerosols and the mesoscale to synoptic-scale meteorology (Vinoj et al., 2014;Ram et al., 2010;Kaskaoutis
et al., 2012a), as well as influences of the ENSO (El Niño southern oscillation) on West Asia and the IGP
(Kim et al., 2016). AOD values were enhanced or elevated during the winter, with the aerosol load
building up throughout the pre-monsoon months ($AOD_{500nm}$>0.6, Figure 3a, 3b, and S5) and then falling
to their lowest values in the monsoon months ($AOD_{500nm}$ ~0.2-0.3), most likely due to wet removal of
aerosols. After the low AOD during the monsoon, AOD gradually increases (to ~0.4-0.5) during the post-
monsoon through winter to the pre-monsoon season.  AOD was usually highest in April
($AOD_{500nm}$:0.86±0.36), followed by March, May and June. The increase in aerosols load (as reflected by
the AOD) during the pre-monsoon months can also be seen at high altitude sites such as the NCO-P site
in the Khumbu Valley near Mt. Everest, located at 5057 m (a.s.l.) and about 300 km to the east of
Pokhara (see Figure 3c), as well as at IGP sites in Kanpur (130 m a.s.l., 400 km southwest of the Pokhara
Valley) and Gandi Nagar (60 m a.s.l., 250 km south of the Pokhara Valley). A similar AOD build-up was
also observed by Ram et al. (2010) in Darjeeling (2194 m a.s.l., hill station ~450 km east of Pokhara
Valley), and by Chatterjee et al. (2012) in Manora Peak (1950 m a.s.l., 460 km west of Pokhara Valley).
This regional increase in aerosol load in the IGP and Himalayan region is partly due to active transport



during the pre-monsoon season, often linked with westerly advection bringing dust from West Asia and
nearby arid regions (Gautam et al., 2011). The relatively dryness with little precipitation during this
period also contributes to the total aerosol load, since washout will be limited. The AOD peaks occur in
different months in these different sites in the IGP and Himalayas, reflecting the varying influence of
local meteorology and increase in the emission sources such as agriculture residue burning dominated
by dominated by fine-mode particles (Putero et al., 2014).

Fine-mode aerosol particles scatter more at shorter wavelengths (such as 340-500 nm) compared

to 1020 nm (Schuster et al., 2006). The variation in the Ångstrom exponent was not as definitive as in
the AOD values; the Ångstrom exponent was generally below 1 during pre-monsoon months and above
1 in the post-monsoon and winter months. Ångstrom exponent values of >1 are generally reported for
sources such as biomass burning, fossil fuel combustion and other primary sources which have a
dominant fine-mode fraction. Dust and other coarse-mode aerosols have Ångstrom exponents less than
1 (Eck et al., 1999). The highest values of the Ångstrom exponent (at least >1.2) were observed for the
post-monsoon observation period, presumably due to emissions of primary fine-mode aerosol from
sources such as open burning of agriculture, often reported in tshis season especially to the south and
southeast of Pokhara Valley and in the IGP. In addition to the Ångstrom exponent, the temporal
variation of AOD fine and coarse modes (at 500 nm) in Figure 3b and 3c also indicates that fine-mode
aerosols nearly exclusively dominate the atmospheric column during the post-monsoon and winter
seasons. In the pre-monsoon season, in addition to the fine-mode, a substantial fraction of coarse-
mode also exists, which is also observed in the monsoon season.

On the nature of aerosols or bulk composition, Figure 3e shows a simple scatter-plot based on the

absorption and extinction Ångstrom exponents (AAE and EAE at 440-870 nm) which can be used to
indicate the aerosol types (Giles et al., 2011;Giles et al., 2012;Dubovik et al., 2002).These two
parameters describe the spectral dependence or "slope" of aerosols absorption and extinction at the
measured wavelength (Seinfeld and Pandis, 2006). Extinction exponent is a proxy for aerosol size, while
the absorption exponent is a proxy for absorbing aerosols including a mixed aerosol. The classification
employed by Giles et al. (2011) based on observations from the IGP AERONET sites defines *"Dust"* or
"*Mostly Dust*" aerosols within the range of EAE <0.5 and AAE >2.0 and "*Mostly BC like"* aerosols with
EAE EAE <0.8 and AAE ~1.0-2.0. Urban/industrial and biomass burning aerosols fall under the "Mostly
BC" category (Dubovik et al., 2002;Giles et al., 2011). The mixed aerosol ("Dust+BC") centers around a
value of EAE ~0.5 and AAE~1.5. Based on this approximate classification from a monthly data, the



dominant aerosol in the Pokhara Valley is mostly *BC like*; however, the daily  aerosol characteristics can
vary from more mixed to dust-like in the pre-monsoon months, to more BC-like in the post-monsoon
and winter months.
**Figure 3.** AERONET-based aerosol optical depth and radiative properties in the Pokhara Valley from 2010
to 2016. Monthly summaries are presented using level 2 collections and supplemented with level 1.5 for
missing data points; **(3a)** AOD at seven wavelengths; **(3b)** Inversion products such as fine AOD (AOD-F),
coarse AOD (AOD-C), and total AOD (AOD-T), along with Ångstrom exponent (440-870 nm, magenta
line); **(3c)** AOD-T for Kanpur, Gandi Nagar (both IGP sites in India) and the NCO-P site (labeled EVK2-CNR,
a high altitude site in the Khumbu Valley at the base of Mt. Everest); **(3d)** Seasonal average of volume
particle size distribution grouped by four seasons (the error bar indicates the standard deviation, and
the uncertainty in the calculated size distribution is close to 20 % in the range 0.2 µm <$D_p$< 14 µm). The
four seasons are classified as winter (DJF: December, January and February), monsoon (JJAS: June, July,
August and September), pre-monsoon (MAM: March, April and May) and post-monsoon (ON: October
and November); **(3e)** absorption Ångstrom exponent (440-870) and extinction Ångstrom exponent (440-
870 nm), color-coded for the four seasons

3.2.    Vertical profiles of absorbing aerosols, particle number and size distribution, temperature, and
dew point
The five test flights are labelled as F1-5 in Figure 4, except F3 which is shown in supplement (Fig.

S10). Due to limitations of the flight permit, the test flights were conducted remaining within the
Pokhara Valley as indicated by Figure 1. Among the five flights, F1, F3 and F5 were morning flights and
F2 and F4 were afternoon flights.
**Figure 4.**  Vertical profiles of aerosol species and meteorological parameters during the 5-7$^{th}$ May 2016
test flights in the Pokhara Valley using the IKARUS microlight aircraft. The subplot in each row is
arranged by (**i**) size distribution measured by the Grimm OPC 1.108 (0.3-20 µm), limited to 1 µm in the
figure, (**ii**) Total particle concentration (also indicated as ***TPC***,  Dp >11 nm) measured by the CPC 3760,
along with absorbing aerosol mass density at 370 nm and 880 nm (**iii**) temperature (red line, in °C) and
dew point (black dots, in °C) and relative humidity (or RH %), (**iv**) calculated absorption Ångstrom
exponent averaged for every 500 meters elevation band. For the size distribution plot, the x-axis
represents the optical diameter of the aerosol (nm), and the color bar represents the concentration ($10^x$
in #cm$^{-3}$). Of the five test flights, only F1-2, F4-5 is shown here, F3 is in the supplementary. Number size



distribution data from Flight F3 is not available due to the failure of the Grimm's pump during flight
initiation. In each subplot, the y-axis is the altitude above the mean sea level (in m).  The origin of the y-
axis is at 815 m (a.s.l.).

All the vertical profiles of total particle concentration (also indicated as **TPC** in Figure 4) showed a

strong gradient below 2000 m (a.s.l.). Because of the valley geography, with surrounding mountains of
about ~2000 m (a.s.l.) or higher, it is likely that the gradient observed below 2000 m (a.s.l.) is mainly
caused by the primary emissions in the Pokhara Valley. The number size distribution of accumulation
mode aerosols (Dia. = 0.3 to 0.5 μm) and the total particle concentration (>11 nm) vary similarly as a
function of the altitude. Concentrations near the surface (~<1000 m a.s.l.) were generally higher than in
the elevated air. For example, the total particle concentrations near the surface were mostly >$10^3$ cm$^{-3}$,
but could reach ~ $3 \times 10^4$ cm$^{-3}$ or higher (see F5 in Figure 4) which is  attributed to the coupling of the
shallow boundary layer and the primary emissions in the contained valley topography (Mues et al.,
2017). In contrast, for the afternoon flight F2 (5$^{th}$ May), the concentrations of accumulation-mode
aerosols at 2500-3000 m (a.s.l.) were higher or comparable (OPC size distribution) to the concentrations
observed at ~1000 m (a.s.l.). The total particle concentration at 2500-3000 m (a.s.l.) in F2 also indicated
a polluted air mass (5-$6 \times 10^3$ cm$^{-3}$), clearly elevated compared to the concentrations above and below,
but still notably lower than the concentration at the surface (~$1 \times 10^4$ cm$^{-3}$). The morning profile for the
same day (F1) showed a polluted layer above 3000-3500 m (a.s.l.), slightly higher in elevation than the
elevated polluted air observed in F2, but lower in particle number concentration (3-$4 \times 10^3$ cm$^{-3}$). The
elevated polluted air mass in F1 and F2 could be an indication of transport related to the mountain
valley winds and/or synoptic transport related to the westerlies, common during this season (Gautam et
al., 2011;Raatikainen et al., 2014;Marcq et al., 2010). Pre-monsoon airborne measurements over the IGP
and near the Himalayan foothills during CALIPEX-2009 found a polluted aerosol layer (2-$4 \times 10^3$ cm$^{-3}$ of
0.13 μm dia. size) below 4 km (a.s.l), attributed to biomass burning observed in this particular season
(Padmakumari et al., 2013).

The movement of the boundary layer during the day clearly influences the evolution of the aerosol

vertical profile in the Pokhara Valley. The shallow boundary layer in the night which continues till the
morning led to the accumulation of aerosols below 2000 m (a.s.l.) in the morning profiles (see the
morning flights F1, F3 and F5) and a strong decrease with altitude. However, among the morning
profiles, substantial variations were observed (between F1 and F5); in F5 there is no polluted layer
above 2000 m (a.s.l.), and overall the observed number concentrations indicate cleaner atmospheric



conditions than the profile in F1. The afternoon profiles (F2 and F4) showed a more relatively mixed
profile up to about 2500-3000 m, decreasing then up to the maximum sampled altitude of just above
4,000 m (a.s.l.). Cloud layers were present at and above 4000 m (a.s.l.) in F4 (also indicated by sharp rise
in RH from ca. 3600 m a.s.l.), which may have led to the scavenging of the aerosol by cloud droplets and
thus the observed drop in the number concentration. Among primary sources in the valley contributing
to the aerosol concentration, open agriculture fires are common during the pre-monsoon season.
Occasional interception of the outflow from agriculture fires around 1500-2000 m (a.s.l.) was observed,
resulting in a sharp rise in the total aerosol concentration. The fire signals are clearly evident in flight F2
(~ 1500 m a.s.l.) and F5 (<2000 m a.s.l.).
The mass concentrations of absorbing aerosols estimated from aerosol absorption measurements
by the aethalometer at wavelengths of 880 nm (or BC) and 370 nm (indicators of the presence of
organic aerosols, often referred to as UVBC) are shown in Figure 4 (ngm$^{-3}$, second column of panels, top
x-axis) along with the total particle concentration (# cm$^{-3}$, bottom x-axis). The similarity in the vertical
concentration gradients of the absorbing aerosol mass concentrations and the aerosol number
concentration above 2000 m (a.s.l.) provides evidence of similar emission sources or origin. The BC
concentration was close to 1 μgm$^{-3}$ up to 4000 m (a.s.l.) during the first two afternoon flights, but was
only about ~0.4 μgm$^{-3}$ during F5. The near surface BC concentrations measured in this study were much
lower than surface BC concentrations measured in the pre-monsoon season (2013) in the Kathmandu
Valley (hourly average: ~5-40 μgm$^{-3}$, Mues et al. (2017)), but comparable to winter measurements
(2004) in Kanpur (1-3 min average: ~1-7 μgm$^{-3}$, Tripathi et al. (2005)). In the same study, winter-time
airborne measurements by Tripathi et al. (2005) observed BC concentrations close to 1 μgm$^{-3}$ up to 2000
m (a.s.l.) and a sharp gradient was observed below 400 m (a.s.l.) most likely due to a shallow boundary
layer in winter.
The absorption at multiple wavelengths was used to calculate the absorption Ångstrom exponent
(AAE), shown in the right-most subplot in each row of Figure 4. AAE was calculated by estimating the
slope of the absorption coefficient ($-\frac{dln(Abs(\lambda))}{dln(\lambda)}$) at the two measured absorption wavelengths (440
and 880 nm, absorption as "Abs" and wavelength as "λ" in the equation). The mass absorption
coefficients (MAC) of 14.5 and 7.77 m$^2$g$^{-1}$, as prescribed by the manufacturer of the aethalometer
(Hansen et al., 1984) for wavelength 440 nm and 880 nm, respectively were used to calculate the
absorption coefficient. The calculated AAE was averaged for each 500 m (a.s.l.), as shown in the figure.
Surface AAE was close to 0.8 to 1.2 which indicates the presence of BC from a mix of sources (biomass





burning and fossil fuel combustion.  A source-diagnostic analysis of C-isotopes of elemental carbon (EC)
in TSP (total suspended particulates) collected in Pokhara during April 2013-March 2014 showed that
biomass burning and fossil fuel combustion contributes nearly 50 % each to the (annual average) EC
concentration (Li et al., 2016). AAE values above surface (>1000 m a.s.l.) varied from 0.5 to 2,  but
mostly falling in the range of 0.9- 1.2, which is typically reported  for mixed to *BC like* aerosols from
urban and industrial emissions (Russell et al., 2010;Yang et al., 2009;Dumka et al., 2014). AAE<1 could
also be indicative of a composite aerosol, where a BC aerosol  (or "core") is coated with absorbing or
non-absorbing aerosols (Gyawali et al., 2009).
**Figure 5**. Aerosol extinction coefficient (at 532 nm) vertical profile (left) and aerosol type
classification based on the CALIPSO level 2 retrieval (right).  Only the CALIPSO overpass over the Pokhara
Valley or nearby locations (such as Kathmandu Valley region, and the region to the west of Pokhara
Valley) is included. The extinction profile is averaged for the region 27-28.5° N latitude, which also
includes the Pokhara Valley. The time is in UTC.
The measured vertical profiles were also complemented with CALIPSO retrievals over the Pokhara
Valley (Figure 5). Level 2 (version 4), cloud and quality screened data were used to generate the
vertically resolved extinction (at 532 nm) and aerosol classification. The CALIPSO satellite had only three
overpasses over the Pokhara Valley between 1 and 10$^{th}$ May 2016 (the extinction profile lines with circle
markers are for the Pokhara Valley). Therefore, the satellite overpasses through nearby regions such as
the Kathmandu Valley region to the east and the region to the west of the Pokhara Valley (denoted by
*WestPV* in Figure 5) were also considered.  The range of extinction values for the Pokhara Valley (0.15-
0.25 km$^{-1}$ especially around 2000-4000 m a.s.l.) were similar to pre-monsoon values (0.15-3 km$^{-1}$)
reported in Nainital (a hilly station located ~2000 m (a.s.l.) in India, and 400 km west of the Pokhara
Valley) and slightly less than Kanpur, a site in the IGP, about 400 km to the southwest of Pokhara
(Dumka et al., 2014). A large extinction (>0.5 km$^{-1}$ ) was observed on 1$^{st}$ May  over the Pokhara Valley at
an altitude of 3-4 km (a.s.l.) which can be attributed to smoke (biomass- related) and polluted dust (a
mixture of dust and biomass smoke or urban pollution) as evident by the aerosol type classification.
Aerosols over the IGP and in the proximity of the Himalayan foothills were mainly "Dust" on 1$^{st}$ May.
Although not conclusive, the 7$^{th}$ May aerosol type classification is markedly different from 1$^{st}$ May with
the absence of dust in the IGP, and absence of polluted dust or smoke over the Pokhara Valley.
**Figure 6**.  HYSPLIT (Hybrid Single Particle Lagrangian Integrated Trajectory) 3 day back trajectories of air
masses arriving at 3 different heights (800 m, 1500 m and 2500 m) from above the ground level (AGL~



815 m a.s.l.) in the Pokhara Valley (28.19° N, 83.98° E) during 5-7[th] May 2016. NCEP GDAS (Global Data
Assimilation System) Reanalysis data with 1°x1° horizontal resolution were used as the input
meteorology.
The measured vertical profiles and available satellite data from MODIS (See Figure S8) and CALIPSO
suggest that the synoptic-scale circulation were changing during the study period. The changing synoptic
circulation also influences the transport of polluted layer into the Pokhara Valley. The regional
meteorology station in the Pokhara Valley reported hazy conditions till 5[th] May (see Figure S6) which
disappeared from 6[th] May onwards followed by clear days with scattered clouds during the daytime and
thunderstorms in the afternoon. The variation in the AOD, AOD-F and Fine Mode Fraction (FMF) from
AERONET (only level 1.5 data were available, see Figure S7) also showed that high turbidity in the
atmospheric column, dominated by fine-mode aerosols before 5[th] May ($AOD_{500nm}$>2.0, FMF >0.9), which
declined sharply after 5[th] May. The variation in the horizontal visibility (or visual range) measured at the
meteorology station in the Pokhara Valley further indicates that the intensity of pollution declined
during the study period, especially starting on 5[th] May 2016.
Three day back trajectories (72 h) were generated using HYSPLIT (Hybrid Single Particle Lagrangian
Integrated Trajectory) for air masses arriving in the Pokhara Valley at 800 m, 1,500 m and 2,500 m from
above ground level (AGL) for the test flight period (see Figure 6). The NCEP GDAS reanalysis data with a
1°x1° horizontal resolution were used as the input meteorology for the trajectories.  The majority of air
masses (especially at 1500 and 2000 m AGL) were westerly. A high resolution (0.0625° horizontal)
simulation of air mass trajectories during the pre-monsoon period over the Himalayas and Tibetan
Plateau region by Lüthi et al. (2015) also identified synoptic-scale transport (as westerly advection
around 500 mb) and a convection-enabled polluted airmass  from the IGP as a major mechanism of
transport in the himalayas. Transport by both mechanisms was coupled with the diurnal expansion of
PBL height in the IGP where the trajectory height was similar to PBL height thus allowing mixing up of
the polluted layer, also observed by Raatikainen et al. (2014) over Gual Pahari (IGP site) and Mukteswor
(Himalayan foothill site).
During the study period, the direction of the trajectories varied as the air masses entered Nepal and
eventually into the Pokhara Valley. On 5[th] and 6[th] May, the air masses (at 1500 and 2000 m AGL) were
mostly northwesterly traversing through northern India and western Nepal before entering the Pokhara
Valley. A shift in the trajectory direction from north westerly to south westerly was observed on 7[th] May,
where the trajectories ware moving through central India and the southern foothills into the Pokhara



Valley. The observed shift in the trajectories at 1500 and 2500 m AGL was modulated by the synoptic-
scale changes in the mid-latitude (over Central Asia) air masses (40-50° N) (Lüthi et al., 2015). The
intrusion (in the form of a trough) of the cold and humid air masses from 40-50° N (see Figure 2) into 20-
30° N occurs during the study period. As the "trough" moves eastward, it shifts the synoptic air mass at
20-30° N from north westerly to south westerly on 7[th] May, prior to which the synoptic air masses were
north westerly. The elevated polluted layer on 5 and 6[th] May (or F1-F4 in Figure 4) could be the result of
this modulation of the westerly. The northwesterly airmass entered Nepal via Northern India, where
MODIS retrievals showed a high aerosol loading (See Figure S8), which could be mainly attributed to the
numerous biomass fire events (See Figure S9) observed in North India.  In addition, numerous forest
fires were also reported in western Nepal during the same period. As the airmass origin shifts to south
westerly on 7[th] May (detected during flight 5), the synoptic air mass bypasses the high AOD loading over
north India and contains the cold and relatively clean air from Central Asia. This resulted in the
disappearance of the polluted layer over 2000 m (a.s.l.) during flight 5.
*4.    Conclusion*

This paper provides an overview of the pre-monsoon airborne measurement carried out with a

microlight aircraft platform in the Pokhara Valley in Nepal, the first-of their-kind airborne aerosol
measurements in the Himalayan foothill region. The objective of the overall airborne campaign in the
Himalayan region was to quantify the vertical distribution of aerosols over a polluted mountain valley
region, as well as to measure the extent of regional transport into the Himalayas. In this paper,
measurements from the test flights during May 2016 are summarized. These mainly include vertical
profiles of aerosol number and size distribution, multi-wavelength aerosol absorption, black carbon,
total particle concentration and meteorological variables. The instrument package, designed for a
microlight sampling was fitted to an IKARUS-C42 microlight aircraft. A total of five test flights were
conducted between 5[th] and 7[th] May 2016, including morning and evening flights for about 1-1.5 h each,
as well as vertical spirals to characterize vertical profiles of aerosols and meteorological parameters

The vertical profiles of aerosol species showed strong gradients along the atmospheric column.

The observed concentration gradient was strongly influenced by the mountain valley boundary layer,
which resulted in a sharp gradient below about 1500-2000 m (a.s.l.). The expansion of the boundary
layer was associated with the differences in the morning and afternoon profiles.  Similar vertical profiles
of BC concentrations and aerosol total particle concentrations provided evidence of common emission
sources or co-located origins. The observed BC concentration near the surface (~ 1000 m a.s.l.) was



much lower than pre-monsoon BC concentrations measured in the Kathmandu Valley, but comparable
to values reported during the winter season in Kanpur in the IGP.  The AAE estimates near the surface,
based on the absorption value, fall in the range of 0.9-1.2, which indicates the presence of *BC like* and
mixed (dust, urban, biomass) aerosols.  An elevated polluted layer was observed at around 3 km (a.s.l.)
over the Pokhara Valley during this study. Characterized by a strong presence of dust in the IGP and
polluted continental airmasses over the Pokhara Valley, the polluted layer could be linked with the
westerly synoptic circulations and regional transport from the IGP and surrounding regions. The
direction of the synoptic transport entering the Himalayan foothills and into Pokhara Valley, however,
was influenced by the Westerlies at mid-latitudes (40-50° N). The extent of transport can be better
quantified with regional airborne measurements along the south-north transect through the region
between the IGP and the Himalayan foothills at high altitudes in the Himalayas, including the Pokhara
Valley.  We will explore the extent of such regional transport in a subsequent publication that will be
primarily based on the airborne measurements in phase II (December 2016- January 2017) in the
Pokhara Valley and surrounding region. The subsequent paper will also characterize the extent of
vertical transport from three different mountain valleys located at different elevations along the south-
north transect.
*Acknowledgements.* The authors would like to thank the Ministry of Population  and Environment, Nepal
(www.mope.gov.np), and the Civil Aviation Authority of Nepal (https://www.caanepal.org.np) for
approving this campaign in Nepal. We are grateful for funding for IASS and for this study from the
German Federal Ministry for Education and Research (BMBF) and the Brandenburg Ministry for Science,
Research and Culture (MWFK). We would also like to thank the NASA DAACs for the data repository of
MODIS, and CALIPSO satellite and as well as NOAA for the meteorology data.  Special thanks to the
NASA AERONET team especially Gupta Giri for operating and maintaining the Pokhara station. The work
was only possible by the support and team-work of the Pokhara Ultralight Company and their
operational staff for the aircraft and air traffic management.

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





**List of Tables**
Table 1. Instrument package deployed in the microlight aircraft

| Parameters | Instruments | Method | Sampling time resolution |
|---|---|---|---|
| 1. Particle size distribution (0.3 - 20 µm) | GRIMM 1.108 | Light scattering | 6 s |
| 2. Total particle concentration (>11 nm) | TSI CPC 3760 | Condensation/light scattering | 1 s |
| 3. Aerosol spectral absorption | Magee AE42 | 7 wavelengths, light attenuation | 2 min |
| 4. Dew point sensor | METEOLABOR, TPS3 | Chilled Mirror | 1 Hz |
| 5. Temperature | Thermocouple | - | 1 Hz |
| 6. Data acquisition system | PC 104+ GPS | --- | --- |
| 7. Power supply | Aircraft battery pack, LiFEPO$_4$ battery | 12 V, >15 AH | |





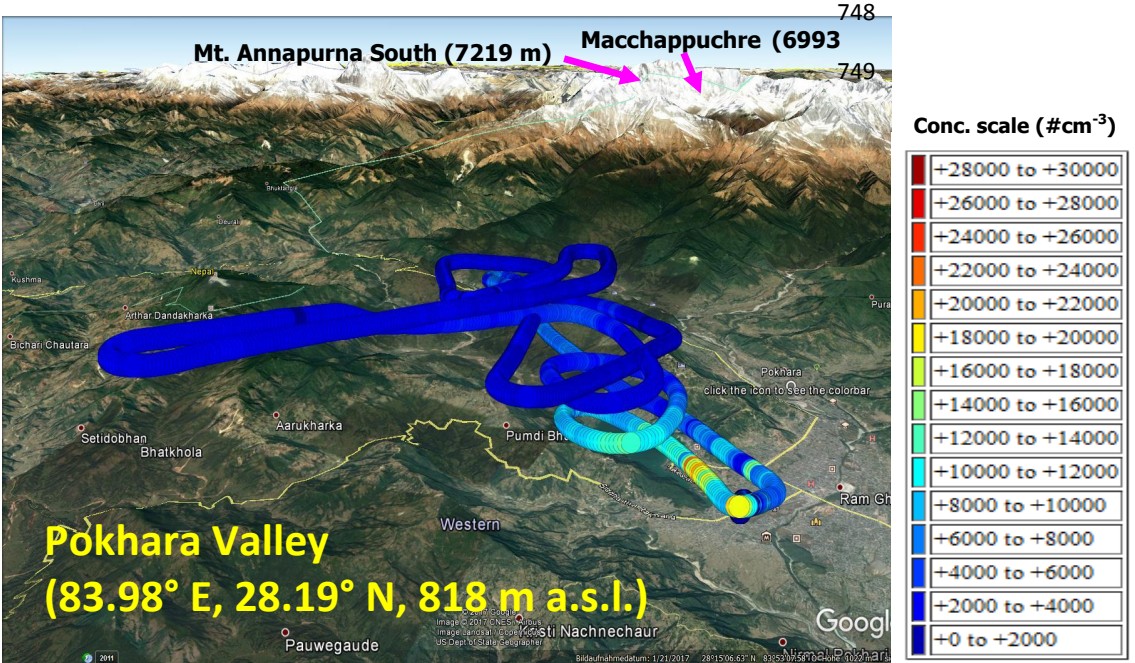


**Figure 1.** A typical test flight within the Pokhara Valley on 5[th] May 2017.  The plot is generated using a Matlab-Google Earth toolbox (https://www.mathworks.com/matlabcentral/fileexchange/12954-google-earth-toolbox). Each dot is a single sample point (sampling frequency of 1Hz) and the color of the dot indicates the total aerosol concentration (in # cm$^{-3}$) and the value of each color is shown as a color bar.





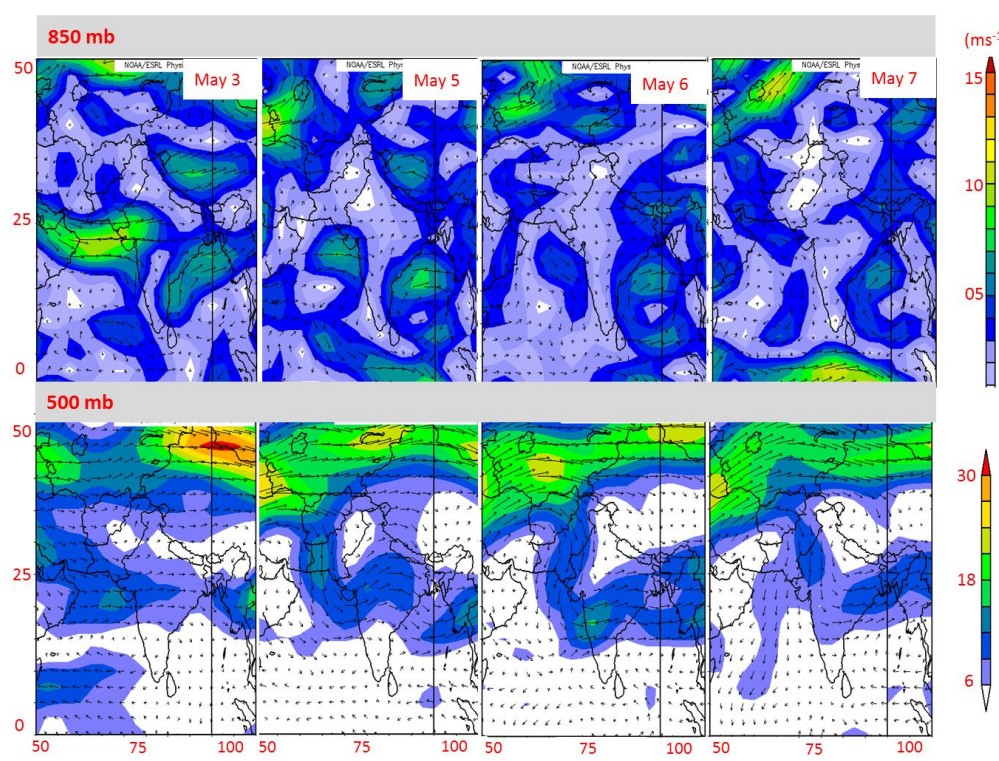

755

**Figure 2.** Daily wind vector data at 850 and 500 mb plotted using the NCEP NCAR reanalysis (2.5° x 2.5°) data over South Asia from 1-7[th] May

2016. The colors indicate the wind speed in ms[-1]. The plots were generated using the default set-up at

www.esrl.noaa.gov/psd/data/composites/day/.



759

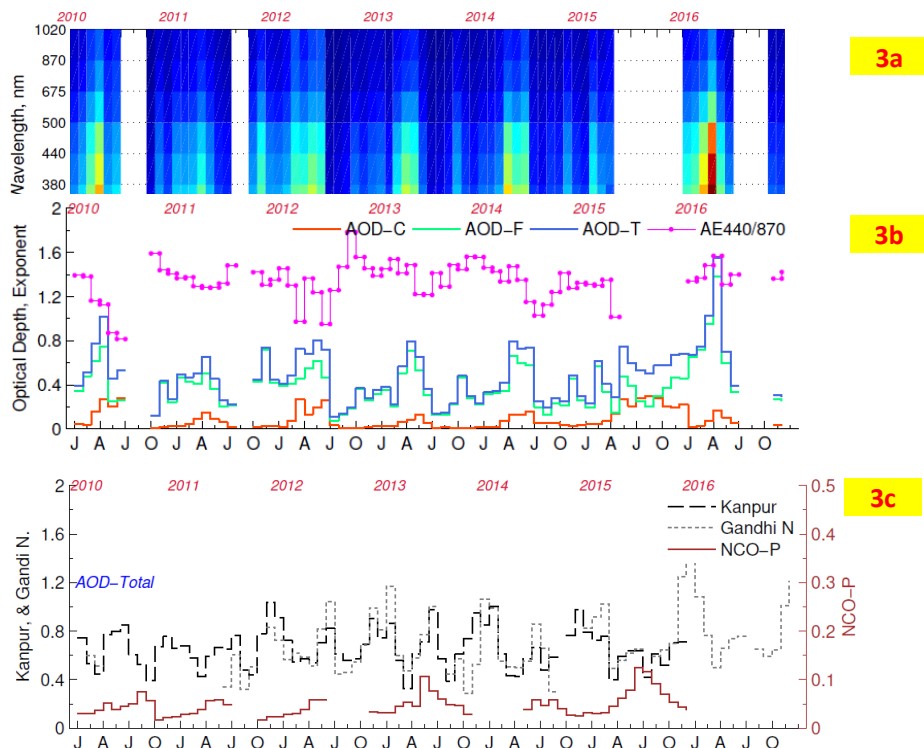

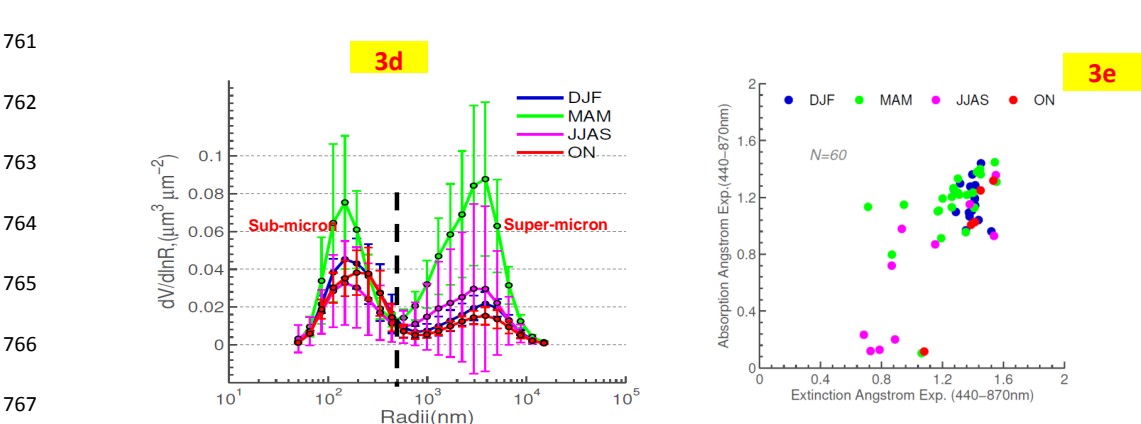

**Figure 3.** AERONET-based aerosol optical and radiative properties in the Pokhara Valley from 2010 to 2016. Monthly summaries are presented using level 2 collections and supplemented with level 1.5 for missing data points; **(3a)** AOD at seven wavelengths; **(3b)** Inversion products such as fine AOD (AOD-F), coarse AOD (AOD-C), and total AOD (AOD-T), along with Ångstrom exponent (440-870 nm, magenta line); **(3c)** AOD-T for Kanpur, Gandi Nagar (both IGP sites in India) and NCO-P site (high altitude site in in the Khumbu Valley in the Himalayas in Nepal); **(3d)** Seasonal average of volume particle size distribution grouped by four seasons (The error bar indicates the standard deviation, and uncertainty in the calculated size distribution is close to 20 % in the range 0.2 μm <$D_p$< 14 μm). The four seasons are classified as winter (DJF: December, January and February), monsoon (JJAS: June, July, August and September), pre-monsoon (MAM: March, April and May) and post-monsoon (ON: October and November); **(3e)** absorption Ångstrom exponent (440-870) and extinction Ångstrom exponent (440-870 nm) grouped by four seasons



779 **Figure 4**

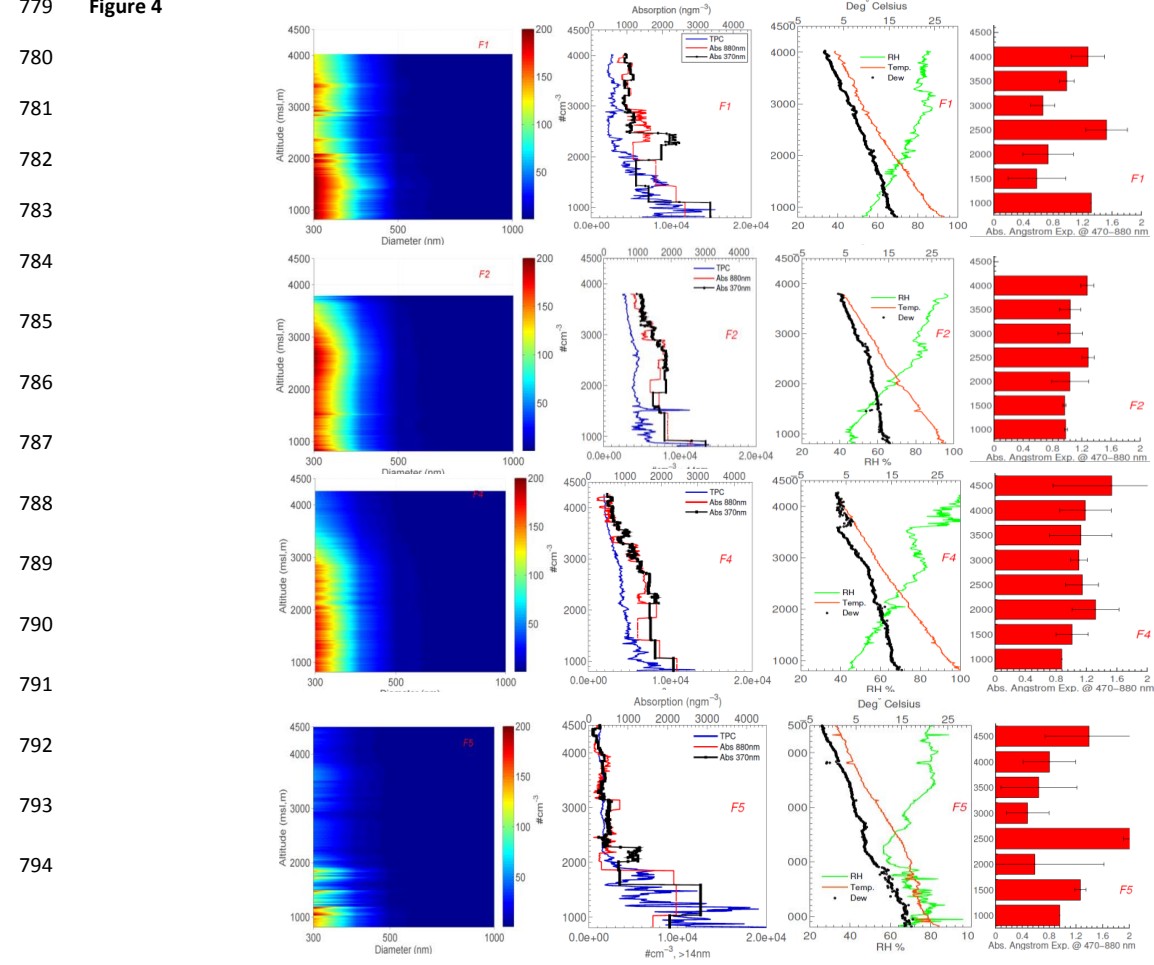



**Figure 4.** Vertical profiles of aerosol species and meteorological parameters during the 5-7[th] May 2016 test flights in the Pokhara Valley using
the IKARUS microlight aircraft. The subplot in each row is arranged by **(i)** size distribution measured by the Grimm OPC 1.108 (0.3-20 µm), limited
to 1 µm in the figure, **(ii)** Total particle concentration (also indicated as *TPC*, $D_p$ >11 nm) measured by the CPC 3760 and absorption aerosol at
370 nm and 880 nm **(iii)** temperature (°C) and dew point (black dot, in °C) and relative humidity (or RH %), **(iv)** calculated absorption Ångstrom
exponent averaged for every 500 meters elevation band. For the size distribution plot, the x-axis represents the optical diameter of the aerosol
(nm), and the color bar represents the concentration ($10^x$ in #cm$^{-3}$). Of the five test flights, only F1-2, F4-5 is shown here, F3 is in the
supplementary. Number size distribution data from Flight F3 is not available due to the failure of the Grimm's pump during flight initiation. In
each subplot, the y-axis is the altitude above the mean sea level (in m).  The origin of the y-axis is at 815 m (a.s.l.).





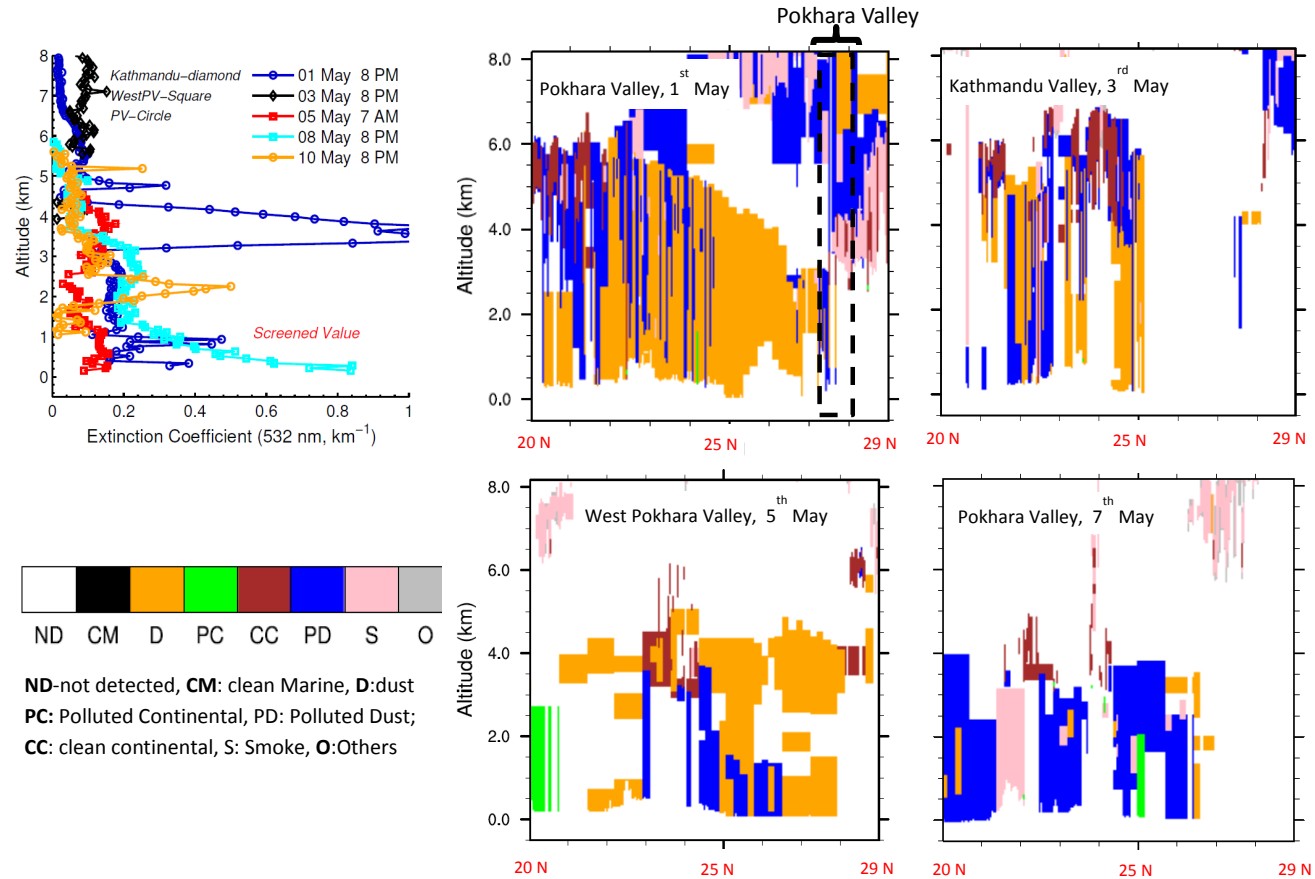





807 **Figure 5**. Aerosol extinction coefficient (at 532 nm) vertical profile (left) and aerosol type classification based on the CALIPSO level 2 retrieval

808 (right).  Only the CALIPSO overpass over the Pokhara Valley or nearby locations (such as Kathmandu Valley region, and the region to west of

809 Pokhara Valley) are included). The extinction profile is averaged for the region 27-28.5° N latitude which also includes the Pokhara Valley. The

810 time is in UTC


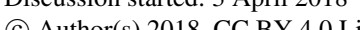



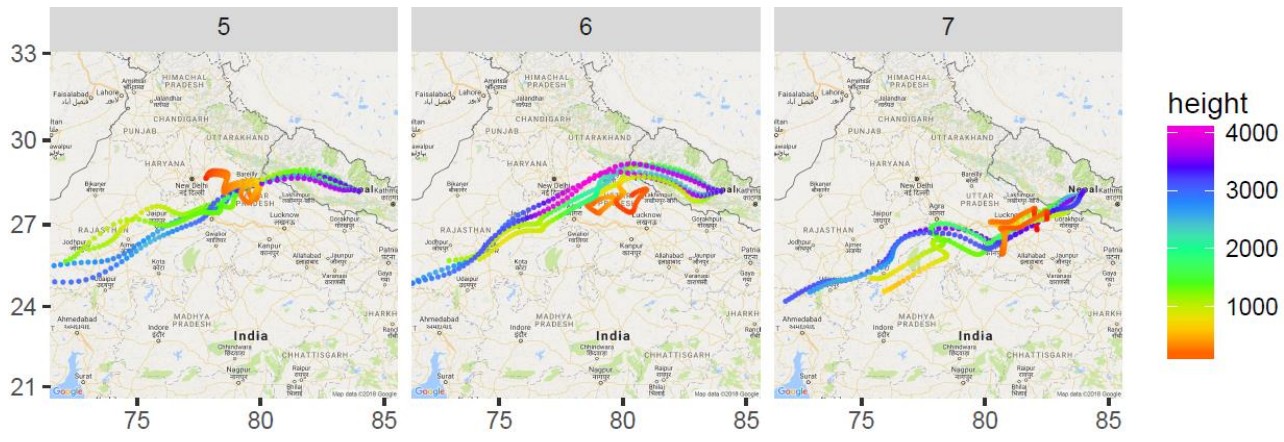


**Figure 6**. HYSPLIT (Hybrid Single Particle Lagrangian Integrated Trajectory) 3 day back trajectories of air masses arriving at 3 different heights

(800 m, 1,500 m and 2,500 m) from above ground level (AGL~ 815 m a.s.l.) in the Pokhara Valley during 5-7[th] May 2016. NCEP GDAS (Global Data

Assimilation System) Reanalysis data with 1°x1° horizontal resolution were used as the input meteorology.

