# Peer review of "An overview on the airborne measurement in Nepal,-part 1: vertical profile of aerosol size-number, spectral absorption and meteorology"

_Atmospheric Chemistry and Physics, 2018_

## Referee Comment (RC1) · Anonymous Referee #2 · 29 Jun 2018

General comments:

This paper present first ever vertically distributed aircraft measurements of atmospheric aerosols over Pokhara Valley in Nepal. The Himalayan region is generally polluted but only few detailed measurements of the pollution exists. Nepal, being situated in Central Himalayas, is a desirable region for conducting such measurements, in order to understand the sources and transport or aerosols in the region. Therefore, the manuscript is of definite value and well suited for ACP. The measurements seem to be well conducted, and complemented with appropriate data from ground based AOD-measurements as well as satellite and model results. There are few issues with the

paper making it lack focus and therefore at times difficult to follow. These issues are mostly minor, and can be addressed with a reasonable effort. I give suggestions below how the authors could improve the paper. I recommend the paper to be published with these minor revisions.

Specific comments:

- Line 63: to add references about new particle formation in the Himalayas, please see Neitola et al. ACP 2011.

- Line 146: which instrument was used to measure meteorological parameters?

- Paragraph 3.1.1. general meteorological features are presented. While these are nice to know, the authors should consider if they give any increased value for the present paper; especially when values outside the measurement period are given. Furthermore, many values are presented without a proper reference to the averaging period (annual, seasonal or for the measurement period?)

- Line 267:" AOD and ground-level PM generally correlate well (Green et al., 2009)". Although the authors soften this statement in the following sentences, I would be very careful to write this, and many people would certainly disagree. Consider replacing the word "well" with something like "to a certain extent". Furthermore, whether association is better with PM2.5 or PM10 is very much dependent on the measurement location, and drawing this conclusion from Illinois (Green et al. 2009 cited) would not prove a wide enough dataset.

- Sub-chapter 3.1.2. I believe this sub-chapter is somewhat too long given the context of the paper, presenting a seasonal analysis of sunphotometer-related products. A somewhat similar analysis and conclusions has been made in the Xu et al. (ACP 2014) paper, although with a shorter measurement period. Similarly to the analysis of the meteorological parameters, the authors should consider whether the analysis brings any additional value to interpret the main measurements (vertical distribution of

aerosols during May 2016) of the paper. Perhaps an analysis focused more on the measurement period could be presented?

- Chapter 3.2. This Chapter presents the core data measured in the project. As written for the moment, the presentation of the results is following a somewhat mixed logic, and I believe some restructuring could be done to make this chapter more readable. First of all, I'm missing the exact dates and times for the 5 flights (F1 to F5) conducted. Second, it would be very useful to identify, if some of the flights were conducted during the same day (as written, F1 and F2 were the morning and evening flights if the same day). Third, I would change the ordering of describing the flights – at the moment for example, the authors write about F2 results before the F1, although it would make more sense to follow the chronological order of the flights, and try to deduce especially what is occurring between flights taking place during the same day.

- At the moment each measured parameter is discussed separately. I would suggest to make a more merged analysis based on the individual flights, in accordance with the previous comment.

- The same Chapter 3.2. includes all the analysis for satellite data and back trajectories. I suggest utilizing more sub-chapters for these.

- Lines 368 – 377. The authors talk about an elevated polluted air mass, first appearing in the morning (Flight F1) at 3000-3500 m a.s.l., and in the afternoon (Flight F2) at 2500-3000 m a.s.l. I am not sure, if these are the same polluted layers. On the contrary, the diurnal evolution of the boundary layer (and / or mountain valley winds during the afternoon) should elevate the aerosol even higher during the afternoon, I would suspect that the morning polluted layer is something else (perhaps long range transported?). The layer which is clearly visible in F2 should then reside at a lower altitude in F1, perhaps around 1500 m a.s.l. Knowing the exact time of the morning flight would give more indication if the polluted layer would already have elevated during this flight.

- Lines 394-396: The authors draw a conclusion that sharp rises in total aerosol concentration at 1500 m a.s.l. during F2 and <2000 m a.s.l. during F5 are due to agriculture fires. I am not sure how this can be deduced from total particle number concentration alone. The aethalometer data should shed some light in the issue, as biomass burning results in an elevated absorption at lower wavelengths compared to 880 nm (i.e. a higher absorption ångström alpha). For the altitudes given by the authors, such elevated absorption at 370 nm is unfortunately not evident. There are some cases where absorption at 370 nm are elevated, namely F1 2000-2500 m a.s.l., F4 2200-2300 m a.s.l., and F5 2000-2300 m a.s.l. These are interesting cases and could warrant more attention.

- Line 399. The authors give the absorption measurement results with ngm-3. This is ok for the 880 nm (typically denoted as Black carbon concentrations). However, for 370 nm, this unit is typically denoted as "apparent black carbon" by the instrument manufacturer. As of, this value does not have any physical meaning, it is rather an indicative measure of absorption in relation to BC. I would encourage the authors to use the absorption coefficient (unit m-1) calculated through MAC for presenting absorption data in the manuscript.

- Line 403: the authors write "during the first two afternoon flights". I thought only two of the flights were conducted during the afternoon?

- Lines 411 forward: the absorption ångström exponent is calculated. Why did the authors choose to take only a two-point slope of the 440 nm and 880 nm measurements, and not a linear regression fit over the whole wavelength range? Both approaches may be used, but I would like to hear their reasoning for this. Further, why did the authors choose to average this data in 500 m bins – was the data too noisy?

- Line 432 onwards: the authors should consider how much the CALIPSO measurement bring added value for the main objective of this paper. Certainly overpasses during 5th and 7th May should be presented with respective overpass times. Are measurements outside the flight days relevant?

[Figure]

- Figure 5: why are the data presented in UTC? This creates much confusion when trying to compare against the vertical flight measurements. Moreover, in the left panel, there is a result for 8th May, while in the right panel for 7th May – where does this discrepancy come from?

- Lines 487-489: the authors conclude that the elevated polluted layer would be due to biomass and forest fires in North India and western Nepal. While this can certainly be one of the reasons, it is unlikely to be the only one, as the biomass burning aerosols should certainly yield higher absorption ångström values (absorb more at lower wavelengths) than observed.

- Supplementary figure S7. This figure is Suphotometer AOD from the flight period, and in my opinion, relevant to be in the main text rather than in the supplementary.

- Figures 6 and S9. Consider overlapping these and presenting in the main text?

Technical comments:

- Line 44: "The intrusions (in the form of a trough) of the cold and humid air mass from the mid-latitude ($\sim$ 40-50° N) a shift in the direction of synoptic airmass entering Himalayas." Unclear sentence, a verb missing?

- Lines 485-486: please remove" prior to which the synoptic air masses were north westerly." As this was already mentioned earlier in the sentence.

- Figure S4: what is the red arrow?

―――――――――――――――――――――――

---

## Referee Comment (RC2) · Anonymous Referee #3 · 20 Aug 2018

This paper describes measurements associated with the first phase of a field campaign aiming to quantify the vertical distribution of aerosol properties over Himalaya and to understand the transport of aerosols to there. The paper is scientifically sound and relatively well written. I have one major comment and a few minor ones that should be addressed before accepting this paper for publication.

My main concern with this paper is its descriptive nature. While the stated goals of the whole field campaign are very valuable, those of the phase 1 are more benign, i.e. mainly to demonstrate that the planned measurement strategy work fine. This is certainly important and worth to be reported properly, but is makes the paper a bit boring

to read. I wonder whether the authors could make section 3 somewhat shorter and more compact. Furthermore, the limitations of this study should be brought up more clearly in abstract and conclusions: five short test flights is enough to demonstrate that the measurement approach works well, but it does not allow making any general conclusions about aerosol sources and properties over Himalaya. The authors need to explicitly tell the readers that the reported observations are just examples of what is going on in that environment.

Other minor comments.

Something (a verb?) is missing from the sentence on lines 45-46 on page at the end of abstract.

The second paragraph on page 10 gives a too optimistic view on the tight relation between AOD and surface PM2.5. They cite to one paper where this correlation is apparently strong, but this is certainly not generally true. This paragraph needs to be rewritten to provide a more realistic connection between AOD and surface PM concentrations.

Figure 3 contains so different panels that, in my opinion, this figure should be split into 2-3 separate figures (3a-3c together making one figure and figures 3d and 3e either combined into one or preferentially separate figures as well).

[Figure]

---

## Author Comment (AC1) · 5 Nov 2018

We thank the reviewer for the review and constructive comments in the manuscripts. In general, significant changes were made in section 3 as advised by the reviewer. The section is relatively concise now, less descriptive and pertains mostly to the flight periods. The changes mainly include less description for section 3.1.1 and 3.1.2; re-organization of flight results in section 3.1.3 and better presentation using multiple sub-headers. Significant changes in section 3.1.3 include removal of Figure 3 into the supplement and replacing Figure 3 with Figure S7 (from the supplement as suggested by the reviewer). Please see the (track change and clean copy) to identify the specific

changes.

- Line 63: to add references about new particle formation in the Himalayas, please see Neitola et al. ACP 2011.

AC: Included. Please see line 59

- Line 146: which instrument was used to measure meteorological parameters?

AC: It is a portable TPS3 model from Meteolabor. The instrument detail is added. Please see line 143

- Paragraph 3.1.1. general meteorological features are presented. While these are nice to know, the authors should consider if they give any increased value for the present paper; especially when values outside the measurement period are given. Furthermore, many values are presented without a proper reference to the averaging period (annual, seasonal or for the measurement period?)

AC: We agree with the reviewer's comment about the relevance of section 3.1.1 for the study. Hence, we have edited (shortened) first 2 paragraphs and included only information which is relevant to the study. However, we have kept the synoptic description as it is. We believe the information provided in the synoptic description is useful and relevant to interpret the results.

Please see the paragraph (line 198 to 209) for the shortened version of the meteorological description and now mainly refers to meteorology during the flight period.

- Line 267:" AOD and ground-level PM generally correlate well (Green et al., 2009)". Although the authors soften this statement in the following sentences, I would be very careful to write this, and many people would certainly disagree. Consider replacing the word "well" with something like "to a certain extent". Furthermore, whether association is better with PM2.5 or PM10 is very much dependent on the measurement location, and drawing this conclusion from Illinois (Green et al. 2009 cited) would not prove a wide enough dataset.

AC: The authors also agree that the relation between AOD and PM is "not so straight-forward" and the citation from Bondville, Iowa may not be appropriate. We would like to avoid such casual statement especially in the current paper which has a different focus or scope. Since we have made significant changes to Sub-chapter 3.1.2. (in light to reviewer's next comment and also suggested by reviewer #2), most of the descriptive part of 3.1.2 is moved to the supplementary information (See supplementary section S7). Only a short summary is presented in section 3.1.2 (Please see line 241-252). The sentence describing the AOD and PM relation is also removed in the text in the supplementary.

- Sub-chapter 3.1.2. I believe this sub-chapter is somewhat too long given the context of the paper, presenting a seasonal analysis of sunphotometer-related products. A somewhat similar analysis and conclusions has been made in the Xu et al. (ACP 2014) paper, although with a shorter measurement period. Similarly to the analysis of the meteorological parameters, the authors should consider whether the analysis brings any additional value to interpret the main measurements (vertical distribution of C2 aerosols during May 2016) of the paper. Perhaps an analysis focused more on the measurement period could be presented?

AC: Thanks for sharing your concern. We also agree with your comment that section 3.1.2 is too descriptive and is not related to the test flight measurement directly. There-fore we have moved the long descriptive (including Figure 3) into the supplementary information (see section S7 in the supplementary). Only a short synopsis is presented which describes general aerosol properties in the Pokhara Valley (Please see line 241-252).

In addition, we have also included the general aerosol properties, and synoptic meteorology observed specifically during the flight period using AERONET and synoptic readings at the Pokhara regional airport (Please see line 257-267).

As also suggested by reviewer #2, Figure S7 is moved from the supplementary information to the main text replacing Figure 3 (as suggested by the reviewer in a comment later) which is relevant to the description provided from line 257-267.

- Chapter 3.2. This Chapter presents the core data measured in the project. As written for the moment, the presentation of the results is following a somewhat mixed logic, and I believe some restructuring could be done to make this chapter more readable. First of all, I'm missing the exact dates and times for the 5 flights (F1 to F5) conducted. Second, it would be very useful to identify, if some of the flights were conducted during the same day (as written, F1 and F2 were the morning and evening flights if the same day). Third, I would change the ordering of describing the flights – at the moment for example, the authors write about F2 results before the F1, although it would make more sense to follow the chronological order of the flights, and try to deduce especially what is occurring between flights taking place during the same day.

- At the moment each measured parameter is discussed separately. I would suggest to make a more merged analysis based on the individual flights, in accordance with the previous comment. - The same Chapter 3.2. includes all the analysis for satellite data and back trajectories. I suggest utilizing more sub-chapters for these.

AC: Thanks for this critical comment.

First, we provide exact date and times of all the flights in the main text (Please see line 273, 276) and also provide Table T1 in the supplementary

We have tried to provide a merged analysis (by discussing all the measured parameters). I think the organization of information flow now aligns chronologically with the test flights (F1-F5). Further, to organize this chapter, we have also broken down the chapter into 4 sub-chapters which are:

3.2.1. Diurnal variation in the profiles (please see line 289-337) 3.2.2. Nature of absorbing aerosols in the Pokhara Valley (please see line 337-356) 3.2.3. Comparison of satellite-derived vertical profiles over Pokhara Valley with aerial measurements

3.2.4. Role of synoptic circulation in modulating aerosol properties over Pokhara Valley - Lines 368 – 377. The authors talk about an elevated polluted air mass, first appearing in the morning (Flight F1) at 3000-3500 m a.s.l., and in the afternoon (Flight F2) at 2500-3000 m a.s.l. I am not sure, if these are the same polluted layers. On the contrary, the diurnal evolution of the boundary layer (and / or mountain valley winds during the afternoon) should elevate the aerosol even higher during the afternoon, I would suspect that the morning polluted layer is something else (perhaps long range transported?). The layer which is clearly visible in F2 should then reside at a lower altitude in F1, perhaps around 1500 m a.s.l. Knowing the exact time of the morning flight would give more indication if the polluted layer would already have elevated during this flight. AC: We do concede that the elevated polluted layer in F2 could be due to the evolution of the boundary layer or the transport related to the mountain valley winds. Therefore we have corrected our presumption that the elevated layer of F2 is long-range transport. Only flight F1 is indicative of elevated polluted air mass (Please see line 321-326) and we have avoided including F2 as an indication of polluted air mass without any further evidence (Please see line 327-336).

Lines 394-396: The authors draw a conclusion that sharp rises in total aerosol concentration at 1500 m a.s.l. during F2 and <2000 m a.s.l. during F5 are due to agriculture fires. I am not sure how this can be deduced from total particle number concentration alone. The aethalometer data should shed some light in the issue, as biomass burning results in an elevated absorption at lower wavelengths compared to 880 nm (i.e. a higher absorption ångström alpha). For the altitudes given by the authors, such elevated absorption at 370 nm is unfortunately not evident. There are some cases where absorption at 370 nm are elevated, namely F1 2000-2500 m a.s.l., F4 2200-2300 m a.s.l., and F5 2000-2300 m a.s.l. These are interesting cases and could warrant more attention.

AC: Thanks for pointing this weakness in our deduction related to the agriculture burn. We agree that the deduction was not well supported by absorption values (at 370 and

880 nm), and number concentration is not a unique and useful indicator of an emission source. The authors would like to point out the condition at which these measurements were taken and hopefully, the reviewer could get a sense of why we came to that seeming deduction.

Agriculture fire was occasionally observed near the difficult terrain in and around the Pokhara Valley. It wasn't a big fire, so the plume disappeared as it rose higher. The sampling team (including the first author) tried to fly as close to the terrain to capture the plume (challenging flight safety sometimes). Fly through the plume was possible (for few seconds), it wasn't possible to hover around it and multiple transects along the plume was also not considered safe. The CPC could capture some pockets of plumes (sample time: 1s) while the aethalometer data (sampling time: 2 min) probably couldn't capture the plume.

Nonetheless, we have removed the speculative deduction from the manuscript. In Section 3.2.1. we do not insinuate anything about the agriculture burn to the spikes seen in the particle number concentration.

- Line 399. The authors give the absorption measurement results with ngm-3. This is ok for the 880 nm (typically denoted as Black carbon concentrations). However, for 370 nm, this unit is typically denoted as "apparent black carbon" by the instrument manufacturer. As of, this value does not have any physical meaning, it is rather an indicative measure of absorption in relation to BC. I would encourage the authors to use the absorption coefficient (unit m-1) calculated through MAC for presenting absorption data in the manuscript.

AC: The absorption coefficient (during the calculation of AAE) is shown in the unit of m-1 and specifically mention it in line 344.

- Line 403: the authors write "during the first two afternoon flights". I thought only two of the flights were conducted during the afternoon?

AC: Sorry, it was a typo. We have edited the sentence. See line 308-309 for the changes.

- Lines 411 forward: the absorption ångström exponent is calculated. Why did the authors choose to take only a two-point slope of the 440 nm and 880 nm measurements, and not a linear regression fit over the whole wavelength range? Both approaches may be used, but I would like to hear their reasoning for this. Further, why did the authors choose to average this data in 500 m bins – was the data too noisy? AC: We looked the spectral distribution of the absorption coefficient (unit m-1) across the wavelength and saw the relation (in log scale) is linear. See the supplement section where we have shown an example plot from the flight data (Figure S11 in the supplementary information; absorption: Y-axis, wavelength: X-axis). Therefore assuming linearity, we thought the two-point slope between 470 and 880 is a good approximation in determining the AAE value. However, to avoid further confusion and objection from the reviewer, we recalculated the AAE value using the regression fit. Please see line 339-342 for the changes in the text related to the calculation of AAE. Also, see the changes in Figure 4 Why 500 m bin? There is no scientific basis for choosing 500 m height resolution to represent the AAE data. The rationale for the using a 500 m was based on the following consideration. 1. Aethalometer sampling rate was sluggish (2 min) which can't provide a 100 m resolution value (also depends on the aircraft vertical climb (120-157 meters/min). 2. In some flights, especially in the lower elevation (see figure), we would have one data point for every 100-200 meters

However, we have presented the data at 100 m height resolution. Please see Figure 4 for the changes

Line 432 onwards: the authors should consider how much the CALIPSO measurement bring added value for the main objective of this paper. Certainly overpasses during 5th and 7th May should be presented with respective overpass times. Are measurements outside the flight days relevant?

[Figure]

AC: The rationale for using CALIPSO was: 1. To compare the profiles generated by aerial measurement to satellite retrieval values (in this case, extinction value) 2. To highlight the changes in the regional air quality using a combination of satellite, and meteorology.

Specific to CALIPSO, we think the CALIPSO data outside the flight days are also indicative of a regional haze or haze condition or high pollution episodes occurring in the broader region which was also indicated by the synoptic meteorology in Figure 3 (previously Figure S7), and in Figure S6. MODIS AOD (in Figure S8 in the supplementary) also showed the high AOD in the region prior to 6th May and a relatively cleaner atmosphere from 6th May onwards.

We have made some changes to the CALIPSO as requested by the reviewer. For instance, we have removed one extinction profiles (only 4 extinction profiles are shown in the figure and they are in local time, not in UTC time). Also, see the changes in Figure 5.

- Figure 5: why are the data presented in UTC? This creates much confusion when trying to compare against the vertical flight measurements. Moreover, in the left panel, there is a result for 8th May, while in the right panel for 7th May – where does this discrepancy come from?

AC: Sorry about the discrepancy. Now the figure is in local time (Nepal standard time). Now the discrepancy is fixed. See the changes in Figure 5.

- Lines 487-489: the authors conclude that the elevated polluted layer would be due to biomass and forest fires in North India and western Nepal. While this can certainly be one of the reasons, it is unlikely to be the only one, as the biomass burning aerosols should certainly yield higher absorption ångström values (absorb more at lower wavelengths) than observed.

AC: Thanks for pointing this out. We think we never tried to be certain about the linkage

between the observed elevated polluted and the biomass and forest fires, it was more of a speculative argument. There is certainly a dust transport along with biomass forest fires (as indicated by CALIPSO).

We have included a few sentences stressing that the biomass burning aerosols could be one of the many other reasons. Please see line 426-429 included to stress this concern.

- Supplementary figure S7. This figure is Suphotometer AOD from the flight period, and in my opinion, relevant to be in the main text rather than in the supplementary. - Figures 6 and S9. Consider overlapping these and presenting in the main text?

AC: We moved Figure S7 to Figure 3. The previous figure 3 is in supplementary as Figure S7. Figure 6 (trajectory data) is merged with the active fire data from Modis C6 collection (Figure S9) and the revised Figure 6 now contains trajectory and active fire count (green+gray circles) for each of the flight days. The green dots are represented by fire radiative powers (frp values) which are in megawatts in Figure 6.

Technical comments: - Line 44: "The intrusions (in the form of a trough) of the cold and humid air mass from the mid-latitude (_ 40-50_ N) a shift in the direction of synoptic airmass entering Himalayas." Unclear sentence, a verb missing?

AC: Corrected. In order to make the abstract short and concise, we have removed the sentence.

- Lines 485-486: please remove" prior to which the synoptic air masses were north westerly." As this was already mentioned earlier in the sentence AC: Corrected.

Figure S4: what is the red arrow?

AC: Removed.

Please also note the supplement to this comment:
https://www.atmos-chem-phys-discuss.net/acp-2018-95/acp-2018-95-AC1-

supplement.pdf

**An overview on the airborne measurement in Nepal,-part 1: vertical profile of aerosol size-number, spectral absorption and meteorology**

Ashish Singh[1]*, Khadak S. Mahata[1], Maheswar Ruphaketi[1]*, Wolfgang Junkermann[2], Arnico K. Panday[3], Mark G. Lawrence[1]

[1]Institute for Advanced Sustainability Studies, Potsdam, Germany

[2]Institute of Meteorology and Climate Research, IMK-IFU, Garmisch-Partenkirchen, Germany

[3]International Centre for Integrated Mountain Development (ICIMOD), Lalitpur, Nepal

*Corresponding author: Ashish Singh (ashish.singh@iass-potsdam.de) and Maheswar Rupakheti (maheswar.rupakheti@iass-potsdam.de)

**Fig. 1.** Final_response_to_RC1

1      **An overview on the airborne measurement in Nepal,-part 1: vertical profile of aerosol size-number,**

2      **spectral absorption, and meteorology**

4      Ashish Singh[1]*, Khadak S. Mahata[1], Maheswar Ruphaketi[1]*, Wolfgang Junkermann[2], Arnico K. Panday[3],

5      Mark G. Lawrence[1]

6      [1]Institute for Advanced Sustainability Studies, Potsdam, Germany

7      [2]Institute of Meteorology and Climate Research, IMK-IFU, Garmisch-Partenkirchen, Germany

8      [3]International Centre for Integrated Mountain Development (ICIMOD), Lalitpur, Nepal

10      *Corresponding author: Ashish Singh (ashish.singh@iass-potsdam.de) and

11      Maheswar Rupakheti (maheswar.rupakheti@iass-potsdam.de)

**Fig. 2.** Revised_manuscript

---

## Author Comment (AC2) · 5 Nov 2018

AC: We kindly appreciate the reviewer's concern about the scope of the paper. We received similar concern from the other reviewer too. Therefore we have tried to address your concerns (for eg. descriptive nature of the paper) by making significant changes in section 3. These changes include:

o 3.1.1 only describes local and synoptic meteorology during flight day period, no overview or other irrelevant description (outside the flight period) were removed. Please see the paragraph (line 198 to 209) o 3.1.2 is shortened to a brief summary (a couple of paragraphs); the original text from the 3.1.2 is moved to the supplemen-

tary as section S7; instead section 3.1.2 now has a brief summary of aerosol properties in the Pokhara Valley and presents observed aerosol properties (using AERONET) and synoptic meteorology data observed during the flight days. Figure S7 replaces Figure 3 which is relevant for the description (and suggested by reviewer #1) and the original Figure 3 is moved into supplement (as suggested). (Please see line 241-252) and line 257-267). o 3.1.3. has multiple sub-headers for better organization of the content and edited to make it more concise and easy read. o Some other minor changes include correction related to the description of the elevated layer, errors related to linking measurement signal with sources, description of absorption AE etc.

Abstract and conclusion also address the reviewer's concern on the limitation of the study (see line 26-29 in the abstract and line 445-447). We have specifically highlighted the limitation of the current study and that the study and that the duration of the study may not be reflective of the air quality, aerosol, and meteorology interaction in the Pokhara Valley

Other minor comments.

Something (a verb?) is missing from the sentence on lines 45-46 on the page at the end of abstract.

AC: Corrected. To make the abstract simple, we have removed the whole sentence.

The second paragraph on page 10 gives a too optimistic view on the tight relation between AOD and surface PM2.5. They cite to one paper where this correlation is apparently strong, but this is certainly not generally true. This paragraph needs to be rewritten to provide a more realistic connection between AOD and surface PM concentrations.

AC: Thanks for sharing your concern. In light with the significant revision requested by the other referee as well, we have reduced section 3.1.2 to a couple of paragraphs. Most of the description in section 3.1.2 is moved to supplement including Figure 3. The

new Figure 3 is Figure S7 from the supplement (as suggested by the other reviewer).

However, we agree that the sentence describing the relation between surface PM and AOD is presented without enough evidence or citation and falls shorts to describe the "not so straightforward" relation that the observation elsewhere has indicated. We have removed that the contentious sentence in the supplementary (see section S7 in the supplementary).

Figure 3 contains so different panels that, in my opinion, this figure should be split into 2-3 separate figures (3a-3c together making one figure and figures 3d and 3e either combined into one or preferentially separate figures as well). AC: We have removed the Figure 3 altogether and is now into the supplementary section. Therefore we will keep the figure presentation as it is (for supplementary).

Please also note the supplement to this comment:
https://www.atmos-chem-phys-discuss.net/acp-2018-95/acp-2018-95-AC2-supplement.pdf

**Supplement:**

supplementary

**DRAFT – DO NOT CITE OR QUOTE**

**An overview on the airborne measurement in Nepal,-part 1: vertical profile of aerosol size-number, spectral absorption and meteorology**

Ashish Singh[1]*, Khadak S. Mahata[1], Maheswar Ruphaketi[1]*, Wolfgang Junkermann[2], Arnico K. Panday[3], Mark G. Lawrence[1]

[1]Institute for Advanced Sustainability Studies, Potsdam, Germany
[2]Institute of Meteorology and Climate Research, IMK-IFU, Garmisch-Partenkirchen, Germany
[3]International Centre for Integrated Mountain Development (ICIMOD), Lalitpur, Nepal

*Corresponding author: Ashish Singh (ashish.singh@iass-potsdam.de) and
Maheswar Rupakheti (maheswar.rupakheti@iass-potsdam.de)

[Figure]

Figure S1: Testing and assembly of the instrument package inside IKARUS-C42 in the COMCO IKARUS station in Germany (a) field station in Pokhara Valley (b) sketch of the Instrument package (c)

[Figure]

Figure S2: Monthly mean values of key meteorological parameters measured at the Pokhara regional airport *(station ID: 804, 28.1993528, 83.9784028, 820 meters altitude)* using an *ENVIRODATA weather station.* December 2016 is missing due to data availability. The monthly value of solar radiation shown here is 95 percentile of the daily values to reflect the peak insolation values. Wind speed and rainfall intensity are 10x for graphical clarity. Note that rainfall values presented here are not cumulative over a month, rather average of the month. (December data is not available)

The annual mean temperature in the valley was 22° C, with the lowest monthly mean in January (~15° C) and the highest in July (~ 25° C). Rainfall was also highest in the months of August and September (summer monsoon season), followed by relatively dry post-monsoon (October-November) and winter period (December-February). The late pre-monsoon to summer monsoon were also the periods of maximum monthly solar insolation (~900 Wm$^{-2}$) and the insolation is approximately half (~550 Wm$^{-2}$) during the winter

[Figure]

Figure S3: Frequency of wind speed and direction observed in the Pokhara Valley during May 2016

[Figure]

Figure S3.1: Wind speed (ms$^{-1}$) and direction from 5-7 May 2016 measured at the Pokhara Regional Airport meteorological station

[Figure]

Figure S4: Daily temperature and relative humidity at 500mb using the NCEP NCAR reanalysis (2.5x 2.5$^{o}$) data over South Asia from May 1 to 7 2016.

[Figure]

Figure S5: Monthly mean value of AOD 500 nm in Pokhara Valley for 2010-2016 (Note: Level 2 and 1.5 were used)

[Figure]

Figure S6: Local Meteorology in the Pokhara Valley from 1-10 May 2017.

Weather condition is coded by individual number (see NOAA https://www7.ncdc.noaa.gov/CDO/dataproduct for details). Weather condition in the figure is shown as a black square box. Weather condition of 5 indicate Hazy conditions;17-thurderstorm but no precipitation; 3-cloud generally forming; 2-sky unchanged

The sky cover is numerically coded as follows:

CLEAR =1, SCATTERED (1/8 TO 4/8) =2, BROKEN-5/8 TO 7/8 =3, OVERCAST=4, OBSCURED=5, PARTIAL OBSCURATION=6

(Also see NOAA https://www7.ncdc.noaa.gov/CDO/dataproduct for details).

[revised manuscript text omitted]

Figure S10: Morning test flight (Flight F3) on 6 May 2016 is shown here, the rest of the results are already shown in in Figure. Each subplot is arranged by (**i**) size distribution measured by the Grimm OPC 1.108 (0.3-20 μm), limited to 1 μm in the figure, (**ii**) Total particle concentration (also indicated as **TPC**, Dp >11 nm) measured by the CPC 3760 and absorption aerosol at 370 nm and 880 nm (**iii**) temperature (°C) and dew point (black dot, in °C) and relative humidity (or RH %), (**iv**) calculated absorption Ångstrom exponent averaged for every 500 meters elevation band.

**Figure S11: Estimating the AAE value using the power fit and linear fit (left: power fit, right: linear fit)**

[Figure]

[Figure]

Table T1

| Flight # | Flight date | Flight time window | | Measured parameter |
|---|---|---|---|---|
| F1 | 5 May 2016 | 7:00-9:00 | Morning flight | T,RH, total particle count, number-size distribution, BC |
| F2 | 5 May 2016 | 14:00-17 :00 | Afternoon flight | " |
| F3 | 6  May 2016 | 7:00-9:00 | Morning flight | " |
| F4 | 6 May 2016 | 14:00-17 :00 | Afternoon flight | " |
| F5 | 7 May 2016 | 7:00-9:00 | Morning flight | " |

---

## Author Response (AR2)

Co-Editor Decision: Publish subject to minor revisions (review by editor) (22 Nov 2018) by Tuukka Petäjä

Comments to the Author:

Editor comments for Singh et al.

Dear Authors,

You have answered the main concerns of the referees well.

I have one main concern and additional editorial comments as follows:

According to the text it clear that you performed the observations in May 2016. Please verify that your data in Figures are from 2016 and not from 2017 as indicated in few places in the text. Further, please add the year to all the dates throughout the text for clarity.

Authors' Response: We would like to thank the editor for providing useful feedback and identifying errors in the manuscript. Apologizes for so many grammatical errors!

FYI: The changes (in response to your comment) made in the manuscript are in GREEN.

Yes, the campaign was conducted in May 2016 and all the data presented in the manuscript are from 2016. We have identified all the typos in the manuscript (figures, main text) where we have incorrectly mentioned 2017, instead of 2016. Now, the manuscript should not have those typos.

Editorial comments:

Line 88: the highest

Done

Line 93: the highest

Done

Line 101: AOD not defined

Done, see line 101

Line 137: model 1.108

Added model, see 137

Line 138: CPC not defined.

Now defined, see line 138

Line 141: DC not defined (not needed?)

Not critical and it's obvious (we think). But we have included the full form now. See line 142

Line 143: define higher.

We removed higher. The data was collected at 1 s (sampling frequency) during the campaign. See line 143

Line 146: LCD not defined (not needed?)

Not critical or needed, but we have included it now. See line 147

Line 155: Is this the length of the sampling tube?

Yes. Now we have indicated that it is the length of sampling tube (in bracket). See line 157

Line 157: How was the sample flow produced for the different instruments? With one general pump or separate pumps for all the instruments?

No separate or independent pump was used. The sample flow is due to the internal pump of each aerosol instrument. We have added a line to highlight this distinction. See line 158-160

Line 175: A typical flight commenced from the Pokhara Regional Airport (818 m a.s.l.) and we flew 5-10 km …

Corrected, see line 178

Line 179: Year should be 2016?

Yes, corrected, see line 183 (also changed in the figure caption later)

Line 184: was .. received

Corrected, see line 188

Line 187: steps of quality control and quality assurance

Corrected, see line 191-92

Line 201: On a local / regional scale, the winds…

Added, see line 205

Line 246: increased

Corrected, line 250

Line 247: the increase of … What do you mean total AOD? Sum of fine and coarse?

Yes, total AOD is the sum of fine and coarse AOD. This is now indicated by the bracketed comment. See line 251-52)

Line 249: … were also present.

Corrected, line 253

Line 250: Reference is needed for the BC-like classification based on Ångström exponents.

The BC-like comment is from Giles et al. 2012. Its indicated now. See line 254

Line 251: was observed

Corrected, see line 256

Line 254: May 2016?

Yes it is May 2016, NOT 2017. Corrected, see line 259

Line 256: The visibility

Corrected, see line 261

Line 258: The aerosol optical …

Corrected, see line 263

Line 262: decreased below 1

Corrected, see line 267

Line 263: The flight day

Corrected, see line 268

Line 281: total particle number concentration

Added, see line 286

Line 293: How can you separate primary and secondary emissions?

Apologies, yes the term primary is used without any supporting evidence. We have modified the sentence. The intention was to indicate that the gradient observed below 2000 masl could be associated or related to emission from the valley. see line 298

Line 294: influenced

Corrected, see line 299

Line 296: continued

Corrected, see line 301

Line 297: remove "profiles". … and a strong decrease with altitude was observed.

Removed the profiles, and corrected the sentence, see line 302

Line 298: total particle number concentration

Corrected, see line 303

Line 300: How can you separate primary and secondary emissions?

Again, apologies for including the term "primary" without sufficient evidence. We have removed the term "primary". See line 305-306

Line 318: reformulate the last sentence for clarity.

The sentence is edited. See line 323-325

Line 329: decreased then

Corrected, see line 334

Line 336: .. thus explaining the observed decrease in the measured aerosol parameters.

Corrected, see line 340

Line 340: please clarify the sentence about calculation of slope.

The sentence is edited and the power equation describing the relation between absorption coefficient and AAE is also included. See line 343-349

Line 345: the absorption coefficient

Corrected, see line 351

Line 348: differed

Corrected, see line 354

Line 349: showed

Corrected, see line 355

Line 353: the biomass

Corrected, see line 359

Line 354: the AAE values

Corrected, see line 361

Line 381: please add the proper scientific reference to HYSPLIT.

The following reference was added. See line 389

Draxler, R. R., and G. D. Hess, 1998: An overview of the HYSPLIT_4 modeling system for trajectories, dispersion, and deposition. Aust. Meteor. Mag., 47, 295–308

Line 392: influenced

Corrected, see line 401

Line 409: transport of air pollution in…

Corrected, see line 418

Line 410: PBL is not defined.

Now defined, see line 419-20

Line 417: were

Corrected, see line 426

Line 421: occurred. Why is "trough" in quotation marks?

Quotation mark removed now, see line 430

Line 428: shorter and longer wavelengths.

Corrected, see line 437

Line 431: bypassed

Corrected, see line 440

Line 432: contained

Corrected, see line 441

Line 446: The results presented in this paper should be considered as a pilot study mapping out the aerosol concentrations and their interactions with meteorological processes in the Pokhara Valley due to the limited flight time.

Thanks for the kind input. See the change in line 455-457

Line 448: aerosol parameters

Corrected, see line 457

Line 449: The observed total number concentration

Corrected, see line 458

Line 451: The increase of boundary layer height contributed to the differences …

Corrected, see line 460

Line 456: fell in the range

Corrected, see line 465

Line 467: in the surrounding

Added, see line 476

Table 1:

1.    Aerosol particle number size distribution

2.    Total particle number concentration

Thanks for the suggestion. See the change in Table 1, from line 648

Figure 3: Please explain AOD_F and AOD_C. Check the year.

Thanks.  See the changes from line 666-669

Figure 4: aerosol parameters, total particle number concentation

Corrected, see the changes from line 688-695

Figure 6: MODIS in capital letters.

Corrected, see the changes from line 708